# TrajAgent: An LLM-Agent Framework for Trajectory Modeling via Large-and-Small Model Collaboration

**Yuwei Du**,* **Jie Feng**\*†, **Jie Zhao, Yong Li**†
Department of Electronic Engineering, BRNist, Tsinghua University, Beijing, China
{fengjie,liyong07}@tsinghua.edu.cn

## Abstract

Trajectory modeling, which includes research on trajectory data pattern mining and future prediction, has widespread applications in areas such as life services, urban transportation, and public administration. Numerous methods have been proposed to address specific problems within trajectory modeling. However, the heterogeneity of data and the diversity of trajectory tasks make effective and reliable trajectory modeling an important yet highly challenging endeavor, even for domain experts. In this paper, we propose *TrajAgent*, a agent framework powered by large language models (LLMs), designed to facilitate robust and efficient trajectory modeling through automation modeling. This framework leverages and optimizes diverse specialized models to address various trajectory modeling tasks across different datasets effectively. In *TrajAgent*, we first develop *UniEnv*, an execution environment with a unified data and model interface, to support the execution and training of various models. Building on *UniEnv*, we introduce an agentic workflow designed for automatic trajectory modeling across various trajectory tasks and data. Furthermore, we introduce collaborative learning schema between LLM-based agents and small speciallized models, to enhance the performance of the whole framework effectively. Extensive experiments on four tasks using four real-world datasets demonstrate the effectiveness of *TrajAgent* in automated trajectory modeling, achieving a performance improvement of 2.38%-69.91% over baseline methods. The codes and data can be accessed via `https://github.com/tsinghua-fib-lab/TrajAgent`.

## 1 Introduction

With the rapid development of web services and mobile devices [68, 6], large-scale trajectory data, such as check-in data from social network [58], have been collected, greatly facilitating research in trajectory modeling. Trajectory modeling involves the processing, mining and prediction of trajectory data, with widespread applications in urban transportation, location services and public management. The typical areas of trajectory modeling [6, 22] can be classified into five main categories: trajectory representation [21], trajectory classification [29], trajectory prediction [60], trajectory recovery [44], and trajectory generation [51]. Each category encompasses various sub-tasks; for instance, the trajectory prediction task can be further divided into next location prediction task [31], final destination prediction task [66], and travel time estimation task [48], among others. Given the huge value of trajectory modeling in diverse practical applications, various algorithms and models [22] have been proposed to address these tasks, particularly deep learning-based models in recent years. This has facilitated significant advancements in the field, with many tasks achieving a high level of performance.

---

*Equal contribution.
†Corresponding author.

39th Conference on Neural Information Processing Systems (NeurIPS 2025).

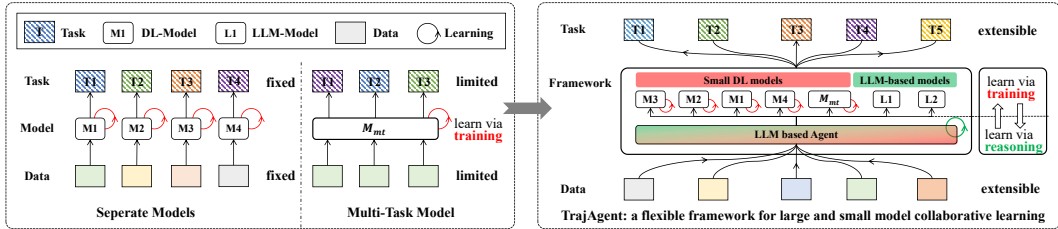

Figure 1: The paradigm of LLM based automated trajectory modeling framework *TrajAgent*.

However, existing methods are designed for specific tasks and datasets, making it difficult to share them across different tasks and data sources. For example, TrajFormer [29] is tailored for trajectory classification and cannot be applied in trajectory prediction or trajectory generation. Flash-back [57] is designed for sparse check-in trajectory prediction and is not suitable for dense GPS trajectory or road network-based trajectory modeling. In other words, due to the heterogeneity of application scenario and the diverse nature of trajectory data–varying in resolution, format and geographical regions– existing methods can only be applied in limited task with specific data for specific regions. While some early studies have explored effective trajectory modeling via unified framework [30, 71, 53], they often face several limitations: 1) their performance on individual tasks lags behind that of specialized models; 2) the range of supported trajectory modeling tasks remains limited; and 3) their training and inference processes are complex and non-trivial. Thus, despite their significant contributions, there is still a long way to go, necessitating further exploration of more effective and reliable trajectory modeling frameworks.

In recent years, the rapid development of large language models (LLMs) [36, 47] with extensive commonsense and powerful reasoning abilities presents enable the LLM based agent [55, 46] as a new paradigm for solving complex task, such as automated software development [23, 38, 59] and automated machine learning tasks [24, 42, 67, 52]. For example, HuggingGPT [42] utilizes LLM as a core manager to address various machine learning tasks with existing AI models, VisionLLM [52] investigates unified modeling for vision tasks across different vision domains. Inspired by these, we explore the potential of leveraging an LLM-based agent framework for automated trajectory modeling, paving the way toward effective and reliable trajectory modeling solution. Specifically, our approach seeks to *harness the capabilities of LLMs to establish a collaborative framework between LLMs and various specialized models, enabling the automated and unified trajectory modeling*. However, designing such an LLM-based agent for this purpose presents several significant challenges. Firstly, how to handle and integrate the diverse trajectory data and specialized models for different trajectory modeling tasks into a single, unified framework is non-trivial. Secondly, the numerous steps involved in transforming raw trajectory data into the final model output are lengthy and cumbersome [6], making full automation of the process difficult and leading to a large action space for both planning and execution. Finally, while model performance heavily depends on delicate and specialized data adaptation and model optimization, the ultimate challenge lies in effectively automating the optimization of these adaptation processes.

In this paper, we propose *TrajAgent*, a systematic agent framework for automated trajectory modeling across diverse tasks and data. First, we design a unified environment, *UniEnv*, to process diverse trajectory data and provide a cohesive runtime environment for various trajectory modeling tasks within *TrajAgent*. In *UniEnv*, we define unified data and model interfaces to facilitate the seamless execution of different trajectory modeling methods. Building upon this environment, we develop an agentic workflow within *TrajAgent* for the automatic multi-step planning and execution of trajectory modeling tasks. The diverse trajectory modeling task workflow is decomposed into four unified steps: task understanding, task planning, task execution, and task summarization. For each step, we design an expert agent to perform the corresponding operations effectively. Finally, we introduce a *collaborative learning schema* that integrates agent learning through reasoning with model learning through training, enabling the effective optimization of model performance for specific data and tasks. We further provide in-depth analysis of optimization dynamics and failure modes, along with practical improvements to enhance robustness. In summary, our contributions are as follows,

- To the best of our knowledge, *TrajAgent* is the first LLM-based agent framework for automated and unified trajectory modeling across diverse data and tasks. It decomposes the trajectory modeling process into several sub-tasks, with expert agents designed to each.

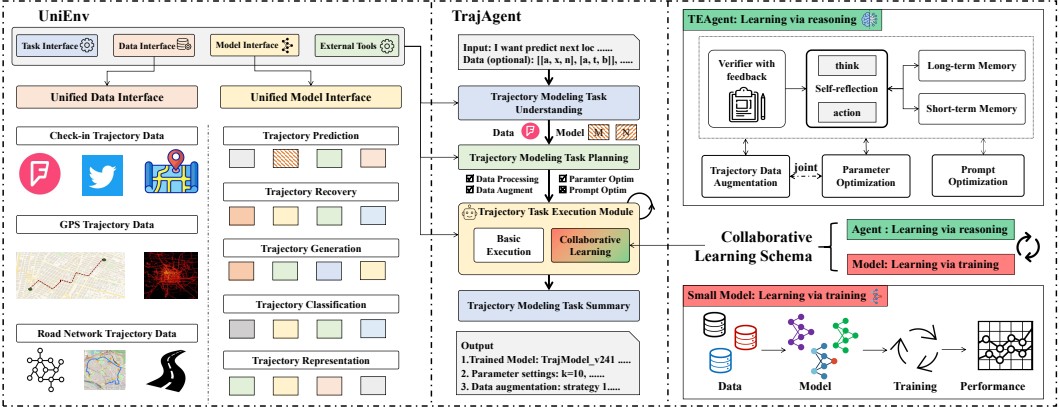

Figure 2: The whole framework of *TrajAgent*.

- To support the automated execution of various trajectory modeling tasks, we provide an unified running environment *UniEnv* by integrating diverse trajectory data and specialized models.
- Furthermore, we develop a *collaborative learning schema* between high-level agents and low-level models to effectively enhance the final performance of *TrajAgent* on targeted data and tasks, proposing a closed-loop, feedback-driven optimization system that jointly adapts data augmentation strategies, model parameters, and agent reasoning based on real-time training outcomes.
- Extensive experiments on four representative trajectory modeling tasks across four real-world datasets demonstrate the effectiveness of the proposed framework, with the optimized model achieving a performance gain of 2.38% to 69.91% over baseline methods.

## 2 Methods

### 2.1 Overview of TrajAgent

Figure 1 presents the comparison between our work and existing works. Over the past few decades, researchers have developed various task-specific models for solving single tasks on limited datasets, as shown in the left of Figure 1. Recently, some works [71, 30, 53] have explored the potential of building unified models for multiple trajectory modeling tasks, as shown in the left of Figure 1. However, limitations in available data and the diversity of tasks have constrained these unified models to a narrow range of tasks. Additionally, these models are not easy to train and utilize, making optimization challenging and far from being simple and user-friendly. Thus, following the success of LLM-based agentic framework in other domains, we propose to build a unify framework to enable the automated trajectory modeling via the collaboration between LLM-based agents and smaller specialized models. The framework is presented in the right of Figure 1. In this framework, the LLM serves as a controller and processor, coordinating with specific smaller models to accomplish specific trajectory tasks. This approach enables users to effortlessly extend support for various data and models without delving into the intricate details of individual models. In essence, *users can view the entire framework as a unified modeling platform, achieved through automated trajectory modeling across a variety of tasks and data.* As TrajAgent *A* is to generate an optimized models *M'*, based on a user query *q*, optional trajectory data *D*, and the selected raw model *M*, the task definition is as follows,

$$M' = \arg\min_M \mathcal{L}(M, T, q, D, A),$$
$$\text{where } T = A(q), D = A(q, T), M = A(T, D). \tag{1}$$

Figure 2 presents the whole framework of *TrajAgent*. It contains three key components: (1) *UniEnv*, an environment with a unified data and model interface that supports the execution and training of various trajectory models; 2) *Agentic Workflow*, which is designed to automatically decompose and complete diverse trajectory modelling tasks; 3) *Collaborative Learning Schema*, an additional automated optimization module for enhancing the performance of specific model through the collaboration between LLM-based agents and specialized models with different learning mechanisms. Details of each component are presented as follows.

## 2.2 UniEnv: Environment for Experiments

As shown in the left of Figure 2, *UniEnv* is a comprehensive and integrated environment that bridges trajectory data, tasks, and models, providing a foundational platform for trajectory modelling and analysis. It is designed to support the entire lifecycle of trajectory modelling workflows, from data preparation to task execution and model optimization. *UniEnv* comprises four key components: a rich set of *datasets* accompanied by processing tools, a comprehensive *task collection* that defines and manages various task types, a extensive *model library* with available source code, and an external *tools pool* for extending the capabilities of *TrajAgent*. Each component is seamlessly connected through a unified interface, enabling agents to plan and execute trajectory modeling tasks with minimal complexity.

**Task Interface:** Figure 3 summarizes the trajectory modeling tasks and associated models supported by *UniEnv*. The framework covers 5 fundamental trajectory modeling tasks: *prediction, recovery, classification, generation, and representation*, utilizing a total of 18 methods. For tasks such as prediction, recovery, and classification, we further introduce several subtasks. For example, the prediction task includes subtasks like next location prediction and travel time estimation, the classification task includes subtasks like trajectory user linking, intention prediction and anomaly detection. To enhance the clarity and effectiveness of understanding users' language queries, we provide a detailed language description for each task. This description helps extract precise task requirements from user queries and facilitates subsequent data and model selection processes.

**Data Interface:** *UniEnv* supports two commonly used trajectory data formats, namely Checkin trajectory (i.e., sequence of visited POIs) and GPS trajectory (i.e., sequence of gps points). These datasets, which come from different cities and with distinct forms, are pre-processing through a standard pipeline that ensures compatibility across the system. Pre-processing steps are done by generated code scripts from LLMs, include data cleaning, normalization, format transformation, which are crucial for handling inconsistencies between real-world datasets and task models. After processing, we will add a description for each dataset to support efficient data selection in the subsequent stage.

**Model Interface:** As previously mentioned, we support 18 models across 5 fundamental trajectory modeling tasks. To support training these models in *TrajAgent*, we select at least one well-known model for each task and adapt them to match the running environments in *UniEnv*. Furthermore, we extract the semantic context from original paper of each model with the txyz.ai APIs [3] to generate detailed description of model. In this description, the verified data information and supported trajectory task information are recorded which supports the data and model selection in the subsequent trajectory planning stage of *TrajAgent*.

**External Tools:** To extend the capabilities of *TrajAgent*, we collect several external tools in *UniEnv*, including paper context extraction tool txyz.ai, hyperparameter optimization tool optuna, processing and visualization tool for trajectory data movingpandas, open street map data processing tool osmnx. Here, we also regard the LLM APIs utilized in the agentic workflow as one of the interface in the *UniEnv*, including the ChatGPT API and DeepInfra for open-source LLMs.

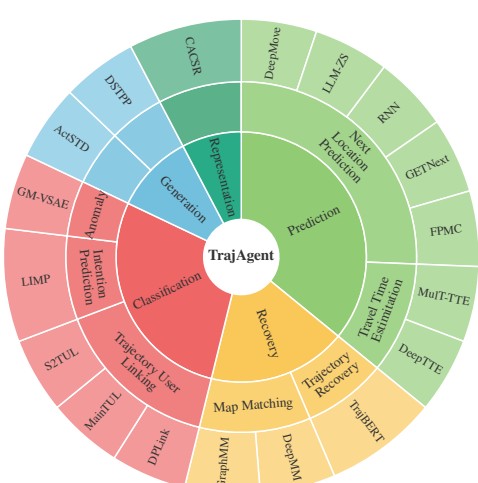

Figure 3: The *TrajAgent* framework supports 5 fundamental trajectory modeling tasks, encompassing a total of 18 methods. Detailed introduction of methods can refer to the appendix A.

## 2.3 Agentic Workflow of TrajAgent

As shown in the middle of Figure 2, the agentic workflow of *TrajAgent* is organized into four key modules: task understanding, task planning, task optimization, and task summary, which form an

---

[3]https://www.txyz.ai/

automated processing chain from user query to final result, eliminating the need for human-in-the-loop. Specifically, the task understanding module first receives user instructions in natural language form, and analyzes and identifies the type, name, and other key information of the tasks involved. Then, the task planning module will plan for the identified task, including dataset matching and model selection. Next, the task execution module executes the planning task and cooperate with the additional performance optimization module *collaborative learning schema* to further improve the task performance from both the agent learning and model learning perspectives. Last, the task summary module generate an analytical report of the task based on historical interactions and decisions of *TrajAgent*. Following the common practice[43], each module in *TrajAgent* can be regarded as a small agent, consisting of a function module for executing its core function, a memory for recording the history interaction, and a reflection module for learning practice from the memory.

**Task Understanding Module:** As the first module of *TrajAgent* workflow, task understanding module is designed to interact with user and extract detailed task information to launch subsequent stages. Given the user query, understanding module recognize the potential task name from it with the predefined supported tasks as additional input. If users ask for the out-of-scope tasks which has not been supported in *UniEnv*, we will directly recommend user to select task from the supported list.

**Task Planning Module:** Follow the task understanding module is the planning module which is designed to generate the subsequent execution plan for efficient experiments of trajectory modelling. The input of the planning module is the task name and description from the understanding module, the supported data and model with brief description from *UniEnv*. With the carefully designed prompt, the generated execution plan will contain the data name and model name for the given task, and also the detailed model optimization plan. Due to the characteristics of different tasks and existing practice, not all the model optimization are necessary for each task. If possible, skipping the optimization step which is time-consuming and costly can accelerate the whole procedure without sacrificing performance. After generating the plan, it will start a simple execution step to verify the feasibility of the plan. Once any error occurs during the execution, e.g., the model name is wrong, the planning module will obtain the feedback from *UniEnv* and start to regenerate a new plan with the last plan as the failed history in its memory.

**Task Execution Module:** Give execution plan, the task execution module is responsible for invoking *UniEnv* to execute the experiment plan. In addition to the previously mentioned basic execution interface, another interface of this module is to call *collaborative learning schema* module to complete the model optimization automatically. For both interface, the task execution module will give the feedback including error information for failed cases and performance metrics for success cases.

**Task Summary Module:** After the execution module, we design a task summary module to analyze the execution records to generate the optimization summary of the task. The summary contains the optimization path during the experiment and the final optimization result for the given task. User can also directly utilize the optimized model from the experiments via APIs for further applications.

## 2.4 Collaborative Learning Schema

Due to the geospatial heterogeneity and the diversity of trajectory data, the trajectory models usually cannot be directly transferred between data and regions. In other words, for various data in different region, the model needs to be trained from scratch. Thus, the sufficient optimization of various models with targeted data becomes emergent. In *TrajAgent*, we design *collaborative learning schema* to complete this automatic optimization and generate optimized specialized models for targeted task and data. As shown in the right part of Figure 2, the collaborative learning framework involves two levels of learning: high-level knowledge reasoning for agents and low-level data training for specialized models. The high-level agent proposes training settings for the low-level models based on its expert knowledge and experimental records. The low-level models are then trained using these settings, and their performance metrics are reported back to the high-level agent for further collaborative learning. This iterative process continues until the performance meets the predefined requirements or the maximum number of exploration epochs is reached.

### 2.4.1 Agent Learning via Reasoning

Following Reflexion [43], we design expert optim agent for learning from the experimental records via reasoning. The standard optim agent utilizes the history operation and related results as the

feedback to update its action in the next step. Specifically, it works as two stages, including "think" and "action". To support the "think then action", it builds a long-term memory for recording all experimental data and a short-term memory for historical actions. In the "think" stage, optim agent analyzes the long-term memory and meta information of experiment and generating the guidance of action in the short-term memory. In the "action" stage, optim agent analyze the results in short-term memory to generate the action. Different optimization mechanism utilize the same optim agent with different action space and optimization tips. Besides, as shown in Figure 1, our framework can also utilize LLM-based agent as the specialized model to complete the trajectory modelling tasks. Thus, we design prompt optimization agent to optimize agent based specialized models. We keep all experimental records in each experiment, including the raw input trajectories, the LLM's inference results, and the reasoning process. We select the two best-performing trajectories along with their corresponding inference results and reasoning process as **high-scoring memory entries**, which are then added to the original prompt for a re-run of the experiment. This process of conducting experiments and selecting results is referred to as one iteration. In each iteration, the **high-scoring memory entries** are updated based on the experiment results.

To validate that agents leverage causal reasoning over memory logs (not pattern matching), we conduct an ablation where memory entries were stripped of performance scores (i.e., no "good/bad" labels). As shown in Appendix Table 13, performance stagnated ($\Delta$Acc@5 < 0.5%), confirming that score-guided reflection is essential. This mirrors reinforcement learning's reward signal, enabling the agent to infer why certain actions succeed.

### 2.4.2 Model Learning via Training

By collaborating with the high-level agent, the low-level specialized models learn specific trajectory patterns tailored to the target trajectory data and tasks. To enhance performance, we introduce several optimization techniques, including data augmentation, parameter tuning, and joint optimization.

*Data Augmentation.* Based on the high-level optim agent, we introduce the specific action space for low-level trajectory data augmentation. For GPS/map-based trajectories, we adopt a geometry-aware augmentation pipeline inspired by DeepMM [12]: (1) raw trajectories are first downsampled to preserve spatial topology; (2) noise is injected only within road network constraints; (3) augmented samples are validated for temporal consistency before merging with original data. For check-in trajectories, we follow the practice from existing works [69, 9, 63], defining a fixed set with ten operators for trajectory data augmentation, e.g., insert, replace, split and so on. During the optimization, the operator set with simple description is provided to the optim agent, it can select optimal operator sequence(combination of operators with their simple parameters) and parameter configuration as the action, guided by training feedback. Then the specialized models are trained with the augmented trajectory data and report performance metrics. The optim agent obtain the feedback information, e.g., performance metrics, from *UniEnv* to continue update its memory and action.

*Parameter Optimization.* The action space of parameter optimization is defined based on the parameters of model itself. We define a parameter configuration file for each model, the optim agent reads the configuration file and generates code as the action to update the parameters in it. To better understand the meaning of each parameter, we add comments for each parameter in the file. This kind of action space is flexible to adapt with any models.

*Joint Optimization.* Furthermore, we introduce the joint optimization mechanisms to further improve the performance. Due to the different working paradigms, the direct combination of two kinds of optimizations is unsuccessful. We designate the optimization order to prioritize data augmentation first, followed by parameter optimization. This means that once the performance of data augmentation stabilizes, the agent proceeds with parameter optimization. This procedure can be repeated a fixed number of times until it meets the stop criteria.

## 3 Experiments

### 3.1 Settings

**Data.** We utilize the widely used Foursquare (FSQ) [58] and Brightkite (BGK) [7] in our framework as the default check-in trajectory data. Porto [27] and Chengdu [8] are integrated in the framework as the default GPS trajectory data. Besides, we use Tencent [34] as the road network based methods

Table 1: Performance of representative methods across five fundamental trajectory modeling tasks. For the ten subtasks, only one model is presented for each. 'DA' represents data augmentation, 'PO' denotes parameter optimization, 'PRO' denotes prompt optimization, 'JO' indicates joint optimization, $\delta$ represents performance improvements.

| Task SubTask | Trajectory Prediction | | | | Trajectory Recovery | | Trajectory Classification | | | Trajectory Generation | Trajectory Representation |
| | Next Loc Pre | | TTE | | Recovery | Map-matching | User Linking | Intent Prediction | Anomaly | | |
| Metric Dataset | Acc@5 FSQ | Acc@5 FSQ | MAE Porto | MAE Tencent | MAE Porto | $Acc_m$ Tencent | Acc@5 FSQ | $Acc_i$ Beijing | AUC Porto | MAE Earthquake | Acc@5 FSQ |
|---|---|---|---|---|---|---|---|---|---|---|---|
| Models | GETNext | LLM-ZS | MulT-TTE | DutyTTE | TrajBERT | GraphMM | S2TUL | LIMP | GM-VSAE | DSTPP | CACSR |
| Origin | 0.3720 | 0.3110 | 163.12 | 190.82 | 42.71 | 0.2014 | 0.5755 | 0.745 | 0.9892 | 0.4611 | 0.31 |
| +DA | 0.3894 | – | – | – | – | 0.3258 | 0.6846 | – | – | – | 0.3369 |
| +PO | 0.3995 | 0.3302 | 128.57 | 179.01 | 27.78 | 0.2427 | 0.757 | – | 0.9899 | 0.3584 | 0.3466 |
| +PRO | – | 0.3225 | – | – | – | – | – | 0.7627 | – | – | – |
| +JO | 0.4002 | 0.3350 | 128.57 | 179.01 | 27.78 | 0.3422 | 0.7802 | 0.7627 | 0.9899 | 0.3584 | 0.3472 |
| $\delta$ | 7.58% | 7.72% | 21.18% | 6.19% | 34.96% | 69.91% | 35.57% | 2.38% | 0 | 22.27% | 12% |

Table 2: Comparison of performance on check-in trajectories for TrajAgent with different configurations, which demonstrate the generalization of *TrajAgent* across different tasks, models and datasets.

| | FSQ | | | | | | BGK | | | | | |
| Task | Next Location Prediction | | | Trajectory User Linking | | | Next Location Prediction | | | Trajectory User Linking | | |
| Model Metrics | RNN | DeepMove Acc@5 | GETNext | MainTUL | DPLink Hit@5 | S2TUL | RNN | DeepMove Acc@5 | GETNext | MainTUL | DPLink Hit@5 | S2TUL |
|---|---|---|---|---|---|---|---|---|---|---|---|---|
| Origin | 0.1795 | 0.3422 | 0.3720 | 0.4871 | 0.7551 | 0.5755 | 0.4422 | 0.5570 | 0.5324 | 0.5908 | 0.8993 | 0.5802 |
| +DA | 0.2667 | 0.4018 | 0.3894 | 0.5973 | 0.7551 | 0.6846 | 0.5416 | 0.5647 | 0.6026 | 0.6836 | 0.9613 | 0.6903 |
| +PO | 0.1795 | 0.3422 | 0.3995 | 0.5691 | 0.7686 | 0.7570 | 0.5022 | 0.6041 | 0.6116 | 0.6683 | 0.9552 | 0.7137 |
| +JO | 0.2717 | 0.4018 | 0.4002 | 0.6121 | 0.8010 | 0.7802 | 0.5470 | 0.6100 | 0.6227 | 0.7145 | 0.9622 | 0.7240 |
| $\delta$ | 51.36% | 17.42% | 7.58% | 25.66% | 6.08% | 35.57% | 23.70% | 9.52% | 16.96% | 20.94% | 6.99% | 24.78% |

to support road network based tasks and Beijing [28] with human labeled intention to support the mobility intention prediction task. Finally, to verify the effectiveness of the whole system, we utilize self-instruct method [50] with 5 seed queries to generate 300 user queries as the experiment input.

**Models.** As shown in Figure 3, our framework supports 18 models spanning 5 core trajectory modeling tasks, which are further categorized into 9 subtasks. The next location prediction task includes FPMC [39], RNN, DeepMove [13], GETNext [60], LLM-ZS [2]. The travel time estimation task comprises DeepTTE [48] and MulT-TTE [32]. The trajectory recovery task features TrajBERT [44], while the map-matching task incorporates DeepMM [14] and GraphMM [34]. The trajectory user linking task includes DPLink [18], MainTUL [5], and S2TUL [10]. The mobility intention prediction task is supported by LIMP [28]. The trajectory anomaly detection task employs GM-VSAE [33], and the trajectory generation task includes ActSTD [65] and DSTPP [64]. Finally, the trajectory representation method is implemented using CASCR [21].

**Metrics.** We adopt standard practices for each task to select appropriate metrics for evaluating our framework. The widely used Acc@k metric is employed for next location prediction, map-matching, trajectory user linking, mobility intention prediction, and trajectory representation tasks. The MAE metric is utilized for travel time estimation, trajectory recovery, and trajectory generation tasks. Lastly, the AUC metric is specifically defined for the trajectory anomaly detection task.

## 3.2 Overall Performance and Generalization Capability of TrajAgent

In this section, we assess the overall performance of *TrajAgent* across various fundamental trajectory modeling tasks, as summarized in Table 1. Furthermore, to demonstrate the generalization capability of *TrajAgent* across different datasets and models, we present detailed results on diverse trajectory data using several representative models in Table 2.

As shown in Table 1, we select at least one representative models for 9 trajectory modelling tasks to present the effectiveness of proposed framework. As Table 1 shows, *TrajAgent* supports a variety of widely-known trajectory models and demonstrates superior performance across multiple trajectory modeling tasks and trajectory datasets. The output of *TrajAgent* consistently outperforms the original methods, achieving performance gains ranging from 2.28% to 69.91%. For instance, in the next-location prediction task, *TrajAgent* harnesses its agentic workflow and collaborative learning schema for automatic modeling and optimization, leading to significant performance improvements of various

Table 3: Execution success rates and task performance at each stage of the agentic workflow of *TrajAgent* across different LLMs. The Acc@5 is obtained by evaluating the next-location prediction task, with DeepMove as the default specialized model.

| LLM | Extraction Succ. | Processing Succ. | Data/Model Selection Succ. | Acc@5 | Data Augmentation Succ. | Acc@5 | Parameter Optim. Succ. | Acc@5 | Joint Optim. Succ. | Acc@5 |
|---|---|---|---|---|---|---|---|---|---|---|
| Qwen2-7B | 85.00% | 30% | 72% | 83.33% | 15% | 0.2015 | 25% | 0.1833 | 64% | 0.2668 |
| Mistral7B-V3 | 78.89% | 42% | 88% | 90.91% | 94% | 0.2940 | **95%** | **0.2087** | 82% | 0.2980 |
| LLama3-8B | 69.44% | 28% | 81% | 80.25% | 18% | 0.1790 | 11% | 0.1809 | 65% | 0.2809 |
| Gemma2-9B | 83.88% | 12% | 57% | 52.63% | 18% | 0.1822 | 15% | 0.1848 | 70% | 0.2970 |
| Gemma-2-27B | 79.44% | 30% | 70% | 88.57% | 78% | 0.2507 | 30% | 0.1775 | 78% | 0.3366 |
| GPT3.5-Turbo | 88.89% | 54% | **100%** | 82.00% | 88% | 0.2846 | 90% | 0.1809 | 92% | 0.3295 |
| LLama3-70B | 83.33% | **100%** | 95% | 86.32% | 92% | 0.2931 | 83% | 0.1848 | 95% | 0.3473 |
| Qwen2-72B | 92.22% | 95% | **100%** | 95% | **96%** | **0.3925** | 70% | 0.1816 | 94% | **0.4333** |
| GPT-4o-mini | **95.56%** | 92% | **100%** | **98.00%** | 90% | 0.2967 | 85% | 0.1822 | **96%** | 0.3724 |

Table 4: Ablation study of *TrajAgent*. 'MS' stands for Model Selection, 'DA' represents Data Augmentation, 'PO' denotes Parameter Optimization, 'JO' stands for Joint Optimization. ↓ indicates a decrease in performance, and ↑ indicates an improvement.

| Agent Variants | MS Succ. | Acc | DA Succ. | Acc | PO Succ. | Acc | JO Succ. | Acc |
|---|---|---|---|---|---|---|---|---|
| TrajAgent | 100% | 98% | 98% | 0.3050 | 89% | 0.1895 | 85% | 0.3724 |
| w/o Reflection | 100% | 95%↓ | 98% | 0.3028↓ | 90%↑ | 0.1872↓ | 82%↓ | 0.3212↓ |
| w/o Memory | 99%↓ | 80%↓ | 85%↓ | 0.1707↓ | 70%↓ | 0.2050↑ | 68%↓ | 0.1804↓ |

models. Specifically, it enhances the deep learning-based model GETNext by 7.58% and the LLM-based model LLM-ZS by 7.72%. Performance gain in other tasks and models are much larger, for example improvement from 34.96% to 69.91% for trajectory recovery tasks. We observe that the improvement in the trajectory anomaly detection task is minimal, primarily due to the strong performance of the original models on the dataset.

As shown in Table 2, we presents a performance comparison for check-in trajectory tasks, including the next-location prediction task and the trajectory user linking task, across three models and two datasets. The first key observation is the consistent improvement observed across tasks, models, and datasets, which highlights the potential generalization of the proposed framework. For different trajectory tasks, datasets, and models, *TrajAgent* consistently provides transferable performance improvements, ranging from 6.08% to 35.57%. Additionally, we observe that the performance gap between different models (e.g., the top two models for each task) on specific tasks and datasets significantly narrows from 24.88% to 9.5% following the automatic optimization of *TrajAgent*, emphasizing the critical role of effective data augmentation and parameter optimization.

### 3.3 Ablation Study and Parameter Analysis of the Agentic Workflow

Here, we select the next location prediction task as an example to demonstrate the efficiency of designs of agentic workflow and the effects of various LLMs in Table 3 and Table 4. During the experiment,we select DeepMove as the default specialized model for next location prediction task.

Table 3 compares the execution efficiency of *TrajAgent* implemented by different LLMs, across each stages in the trajectory modelling workflow. We can observe that Qwen2-72B and GPT-4o-mini demonstrate the highest success rates (i.e., Succ.) across key stages such as Data Extraction, Processing, and Data/Model Selection, with over 90% success in each. In contrast, models like Gemma-2-9B and LLama3-8B struggle with lower processing and data selection success rates, which results in reduced overall performance. Their weaker performance in key optimization stages, especially in parameter selection, reflects their limitations in effectively supporting TrajAgent. These results show that TrajAgent's efficiency is strongly influenced by the base LLM, with high-performing models like Qwen-72B and GPT-4o-mini significantly enhancing its capabilities.

We conducted an ablation study by isolating two main designs: the memory unit and the reflection mechanism, resulting in two variants: 1) **w/o Reflection**, where the reflection mechanism is removed, and 2) **w/o Memory**, where the memory unit is excluded. The experimental results are presented in Table 4. We can find that: 1) Removing either component leads to average performance declines,

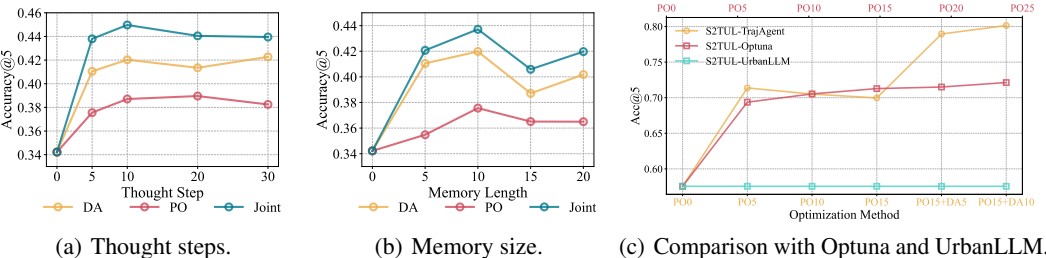

(a) Thought steps.  (b) Memory size.  (c) Comparison with Optuna and UrbanLLM.

Figure 4: (a-b) The impact of thought steps and memory size when using DeepMove in next location prediction tasks. (c) Compared to Optuna and UrbanLLM, TrajAgent achieves better performance in trajectory user linking tasks with S2TUL as the specialized model.

with the memory unit being especially critical for maintaining execution efficiency. 2) Interestingly, removing a component occasionally results in slight increases in success or accuracy, suggesting that some components may introduce overhead or complexity in specific stages. Overall, the combined use of the memory and reflection mechanisms is crucial for optimizing *TrajAgent*'s performance.

Due to the importance of reflection and memory in the *TrajAgent*, we analyze the effects of two important related parameters: 1) **thought step**: the number of steps that agent thought before taking action in reflection, and 2) **memory size**: the size of memory units in *TrajAgent*.

As shown in Figure 4(a), performance initially improves with more thought steps, reflecting enhanced reasoning depth. However, beyond a threshold (e.g., 20 steps), performance declines—likely due to overfitting to suboptimal action sequences or repetitive exploration without sufficient novelty. Similarly, in Figure 4(b), increasing memory size beyond 10 leads to performance degradation, suggesting that excessive historical records may introduce noise or bias the agent toward previously failed strategies (a phenomenon we term the 'optimization trap').

### 3.4 Analysis of Optimization Failure Modes and Improvements

While TrajAgent demonstrates strong performance across diverse trajectory tasks, we observe that its optimization efficacy does not monotonically improve with increasing reasoning depth or memory capacity. To understand this phenomenon and enhance robustness, we conduct an in-depth analysis of the underlying optimization dynamics.

**Optimization Trap in Long Reasoning Chains**: As the number of thought steps increases, the agent may converge prematurely to a local optimum. Once trapped, the agent stops exploring novel strategies and repeatedly refines the same ineffective combination. This behavior is exacerbated in weaker LLMs, which lack sufficient reasoning capacity to escape such traps. In contrast, reasoning models like DeepSeek-v3 exhibit more diverse exploration and are more likely to discover globally superior configurations within fewer steps (see Table 10 in Appendix).

**Memory Saturation and Noise Accumulation**: Similarly, increasing memory size improves exploration efficiency up to a point, but larger memories introduce noisy or redundant historical records. Low-performing action sequences, if retained, can bias future decisions and lead the agent to revisit failed strategies—especially when memory is not actively curated. This "memory pollution" effect explains the performance drop in Figure 4(b). Crucially, stronger reasoning models (e.g., DeepSeek-v3, Gemini-2.0) are less affected, as shown in Table 10 in Appendix: they maintain or even improve performance with larger memories by better filtering useful experiences, whereas weaker models degrade significantly. This highlights memory content management and optimization as a key factor in mitigating memory pollution.

It is also worth noting that although Figure 4(b) shows parameter optimization (PO) is less sensitive to memory size than data augmentation (DA), excessive memory (>10 entries) still degrades PO due to noise accumulation. The apparent stability in PO stems from its smaller action space, but memory pruning is equally critical for both.

**Mitigation via Contrastive Reflection and Memory Pruning**: To address these issues, we introduce two practical improvements: (1) Contrastive Reflection: During the reflection phase, the agent

explicitly compares successful and failed trials, adjusting operator parameters to avoid repeating ineffective combinations. This encourages diverse yet informed exploration. (2) Dynamic Memory Pruning: We periodically discard low-scoring memory entries and retain only high-performing trajectories as high-scoring memory entries to guide future planning. As shown in Tables 12 and 11 in Appendix, these strategies significantly stabilize optimization.Similarly, memory pruning enables consistent gains across memory lengths, with performance no longer collapsing at large capacities.

## 3.5 Comparison with Automated Methods

We compare *TrajAgent* with the deep learning-based AutoML method Optuna [1] and the LLM-based automated urban task-solving framework UrbanLLM [26]. Using the trajectory user linking task with S2TUL model as an example, the results are presented in Figure 4(c). UrbanLLM (represented by the blue line) is designed to directly automate the use of existing models without further optimization, resulting in the lowest performance in Figure 4(c). Meanwhile, compared to the widely used AutoML method Optuna (represented by the red line), which focuses on parameter optimization, *TrajAgent* (depicted by the yellow line) achieves superior results with fewer trial-and-error iterations. Furthermore, its performance can be further enhanced through joint optimization combined with automated data augmentation, resulting in a significant performance improvement of over 11.1%. For computational overhead please of *TrajAgent*, please refer to Table 8 in Appendix.

## 4 Related Work

**Trajectory Modelling and Analytics:** In recent year, trajectory modelling [68, 6, 22] makes great progress on its core research questions, including prediction [31, 13, 57, 41, 60], classification [25, 20, 18, 29, 45, 5], recovery [54, 56] and generation [37, 65, 51]. While these specific methods accelerate the development of trajectory modelling from different aspects, they can only handle one type of task. In other words, the automated and unified model for all the trajectory modelling task is still missing due to the heterogeneity of tasks and trajectory data. In this paper, we propose to utilize the power of LLM and agent to build a unified model framework for diverse trajectory modelling tasks, which can automatically handle various data and modelling tasks without human intervention.

**Large Language Models:** LLMs with extensive commonsense and outstanding reasoning abilities have been widely explored in many domains, such as mathematics [62], question answering [72], and human-machine interaction [47]. Following this direction and motivated by the exploration of LLMs' usability in urban studies [17], researchers have begun developing diverse domain-specific LLMs tailored for urban applications, for example, CityGPT [15], UrbanLLM [26], and UrbanLLaVA [16]. In contrast to these approaches, which rely on fine-tuning LLMs with domain-specific knowledge, our work focuses on a training-free paradigm by constructing an agentic framework.

**LLM based Agents:** In the general domain, agentic framework [49, 46, 55] are proposed to enhance the robustness and task solving abilities of LLMs for real-world complex tasks, such as web-navigation [61, 35, 70] and software development [23, 38, 59]. Besides, researchers also explore the potential of applying LLM based agent for automatic research experiments [3, 40, 4] especially machine learning experiments [24, 42, 67, 52]. Recently, LLM-based agent also be applied in specific trajectory modeling tasks, e.g., trajectory prediction [12] and trajectory simulation [11]. In this paper, our proposed agentic framework is designed for unified trajectory modelling which can provide automatically model training and optimization for various trajectory data and tasks.

## 5 Conclusion

In this paper, we propose *TrajAgent*, an LLM-based agentic framework for automated trajectory modeling. Supported by *UniEnv*, which provides a unified data and model interface, and *collaborative learning schema* for joint performance optimization, *TrajAgent* can automatically identify and train the appropriate model, delivering competitive performance across a range of trajectory modeling tasks. *TrajAgent* establishes a new paradigm for unified trajectory modeling across diverse tasks and datasets.

**Acknowledgments**

This work was supported in part by the National Key Research and Development Program of China under grant 2024YFC3307603, in part by the China Postdoctoral Science Foundation under grant 2024M761670 and GZB20240384, in part by the Tsinghua University Shuimu Scholar Program under grant 2023SM235. This research is supported by Tsinghua University – Mercedes Benz Institute for Sustainable Mobility.

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

## A Appendix

### A.1 Datasets

- **Foursquare (FSQ)**: This dataset consists of 227,428 check-ins collected in New York City from 04/12/2012 to 02/16/2013, Each check-in is associated with a timestamp, GPS coordinates and corresponding venue-category.
- **Brightkite (BGK)**: This dataset contains 4,491,143 check-ins of 58,228 users collected from BrightKite website.
- **Porto**: This dataset contains 1.7 million taxi trajectories of 442 taxis running in Porto, Portugal from 01/07/2013 to 30/06/2014. Each trajectory corresponds to one completed trip record, with fields such as taxiID, timestamp and the sequence of GPS coordinates.
- **Chengdu**: This dataset contains GPS trajectory records of Chengdu from 01/11/2016 to 30/11/2016. Each record includes taxiID, timestamp, longitude and latitude, collected and released by Didi Chuxing.
- **Tencent** [34]: This dataset includes both a road network and a set of vehicle trajectories collected in northeastern Beijing. The road network consists of 8.5K road segments and 15K edges, and the trajectory dataset contains 64K vehicle trajectories, each sampled at 15-second intervals.

- **Beijing** [28]: This dataset contains check-in records collected from a popular social networking platform. It spans a period from late September 2019 to late November 2019.It also contains intent labels annotated by human for a small dataset.
- **UserQueries**: To verify the effectiveness of the whole system, we utilize self-instruct method [50] with 5 seed queries to generate 300 user queries as the experiment input.

More detailed information for the Check-in and GPS trajectory datasets are summarized in Table 5 and Table 6, respectively. Note that the raw datasets are typically city-scale and data-intensive. Loading them all into the TrajAgent framework is costly. In this work, we are selecting a part of data for task model training and testing during experiments.

Table 5: Statistics of the Check-in trajectory datasets.

| Datasets | Foursquare (FSQ) | Brightkite(BGK) | Beijing |
|---|---|---|---|
| Num. Users | 463 | 272 | 1566 |
| Num. POIs | 19870 | 50061 | 5919 |
| Num. Trajectories | 10632 | 22208 | 744813 |

Table 6: Statistics of the GPS trajectory datasets.

| Datasets | Porto | Chengdu | Tencent |
|---|---|---|---|
| Sampling Rate | 15s | 3s | 15s |
| Num. Traj. | 1,710,670 | 5,819,383 | 10,000 |
| Avg. Traj. Length ($m$) | 3522.64 | 2857.81 | 2492.01 |
| Avg. Travel Time ($s$) | 724.20 | 436.12 | 13903.78 |
| Latitude Range | [41.1401, 41.1859] | [30.6529, 30.7277] | [40.0224, 40.0930] |
| Longitude Range | [-8.6902, -8.5491] | [104.042, 104.129] | [116.265, 116.349] |

## A.2 Models

As introduced in Figure 3, we adopt various deep learning based and LLM based models and incorporate them into TrajAgent for solving trajectory-related tasks. According to the type of tasks they deal with, these models can be framed in the following categories:

- **Next Location Prediction**: RNN [31], attention-based method like DeepMove [13], well-performed graph-based method like GETNext [60] and two recent proposed LLM-based methods LLM-ZS [2] and LLMMove [19].
- **Travel Time Estimation**: DeepTTE [48] and MulT-TTE [32] to estimate the travel time for a given GPS trajectory.
- **Trajectory User Linkage**: widely-used DPLink [18], MainTUL [5] and S2TUL [10] are considered. We modified DPLink's training approach by using publicly available sparse trajectory datasets instead of heterogeneous mobility datasets for training.
- **Travel Intent Prediction**: LIMP [28], which leverages the commonsense reasoning capabilities of LLMs for mobility intention inference.
- **Trajectory Anomaly Detection**: we consider the well-performing method GMVSAE [33], which represent different types of normal trajectories in a continuous latent space.
- **Trajectory Generation**: ActSTD [65], which capture the spatiotemporal dynamics underlying trajectories by leveraging neural differential equations and DSTPP [64], which defines spatial temporal point process for trajectory generation.
- **Trajectory Recovery**: TrajBERT [44], which encode trajectory as sentence and train a BERT model to get representations of trajectories.
- **Trajectory Map Matching**: DeepMM [14], which proposes a data augmentation approach for map-based trajectory data, and GraphMM [34], which leverages a graph-based framework to extract features from map-based trajectory data, are considered.
- **Trajectory Representation**: CASCSR [21], which use contrastive learning method to learn trajectory representations for downstream tasks.

### A.3 Metrics

To evaluate the performance of all models on multiple trajectory tasks, we employ the following different metrics:

- **Acc@5** and **Hit@5**: Acc@5 refers to the percentage of the first five results predicted correctly. Hit@5 measures whether at least one of the top-5 predictive results is correct.

- $Acc_m$: $Acc_m$ is the evaluation metric which computes the average matching degree of all trajectories. For each trajectory, its matching degree is the ratio of the number of matching road segments to the number of all road segments.

- $Acc_i$: $Acc_i$ is used to measure the accuracy of trajectory intention inference. It is the ratio of the number of matching predicted intention of each check-in to the total number of check-ins.

- **MAE** and **AUC**: Mean Absolute Error (MAE) indicates the amount of deviation from the actual values. Area Under ROC Curve (AUC) measures how well the model correctly distinguishes the type of the sample.

- **JSD**: Jensen–Shannon divergence (JSD) measures the discrepancy of distributions between the generated data and real-world data. Lower JSD denotes a closer match to the statistical characteristics and thus indicates a better generation result.

In our experiments, Acc@5 is used to measure the accuracy of trajectory prediction and agent execution, it is also used to measure the downstream applications of trajectory representation. $Acc_m$ is designed for measuring the performance of map matching task. $Acc_i$ is designed for measuring the performance of travel intention prediction task. Hit@5 is adopted for evaluating the performance of trajectory user linkage task; MAE is employed to compute the error of travel time estimation and trajectory recovery, while the metric AUC is used to assess the performance of trajectory anomaly detection. JSD and RMSE are used to measure the prediction error of the spatiotemporal domain in trajectory generation task.

All experiments are conducted on a Ubuntu server equipped with 8 NVIDIA GeForce RTX 3090 GPUs. Each small model is trained using a single RTX 3090 GPU, while each large language model (LLM) is accessed through its corresponding API provider.

### A.4 Additional results on GPS trajectory data

Table 7: Comparison of task performance on GPS trajectories for TrajAgent with different configurations.

| Task | | TTE | | Anomaly | Recovery |
|------|------|---------|---------|---------|----------|
| Model | | DeepTTE | MultTTE | GMVSAE | TrajBERT |
| Metrics | | MAE | | AUC | MAE |
| Porto | origin | 8.48 | 129.35 | 0.9892 | 13.6667 |
| | +JO | **5.85** | **109.85** | **0.9899** | **8.0290** |
| | $\delta$ | 31.01% | 15.08% | minor | 41.25% |
| Chendu | Origin | 7.23 | 166.29 | 0.978 | 54.8060 |
| | +JO | **5.95** | **128.57** | **0.984** | **29.8134** |
| | $\delta$ | 17.70% | 22.68% | 0.61% | 54.39% |

Table 7 presents the performance comparison on GPS trajectory tasks, specifically focusing on TTE and TAD tasks. The models evaluated are DeepTTE for TTE and GMVSAE for TAD, with results shown for the original configuration (Origin) and after Joint Optimization (+JO). For the Proto dataset, the original model has an MAE of 8.48, which significantly improves to 5.85 after joint optimization, resulting in a 31% reduction in prediction error. On the Chengdu dataset, the MAE decreases from 7.23 to 5.95. As for TAD task, we find that the AUC scores were already high, the small but consistent improvements in both datasets suggest that TrajAgent's joint optimization can further refine model performance, even for tasks where models initially perform well.

## A.5 Limitations and Failure Mode Analysis

In this section, we analyze some cases where the TrajAgent's optimization performance is suboptimal. The overall process is illustrated in Figure 5. The optimization module is a key component affecting overall performance.

The first issue is *optimization trap* present in data augmentation module of TrajAgent. Specifically, it refers to the situation where agent sometimes ignores the contents of the memory during the thought process. Even when the chosen parameter combination yields poor training results, the model overlooks the error feedback (i.e., the "Not good enough..." in the memory). The "optimization trap" occurs even in the best-performing GPT-4o-mini. As the length of the Memory increases, the impact of the optimization trap on overall accuracy grows. Once an optimization trap occurs, all memories within the same step tend to favor the same ineffective combination. We believe the causes of the optimization trap could be: (1) excessively long prompts leading to truncated inputs; (2) insufficient proportion of memory in the total input.

The second issue is the *sub-optimality* appears in parameter optimization module of TrajAgent. This phenomenon exists across various datasets and model sizes. We believe the reasons for the poor performance might be: (1) the TrajAgent parameter optimization module is overly sensitive to the format of model outputs, treating all responses that do not meet the format requirements as invalid; (2) adjustments to certain parameters result in increased training time, reducing the total number of iterations.

To address these issues, we propose two enhancements: (1) a contrastive reflection mechanism that learns from both successful and failed trials to avoid redundant exploration; and (2) a dynamic memory management strategy that prunes low-performing historical actions. Experimental results (Tables 12 and 11) confirm that these strategies effectively mitigate performance degradation at large step/memory sizes.

## A.6 Additional Experimental Analysis

**The optimization effect diminishes as the step increases**: While selecting combinations of operators, the model further optimizes the configuration of each operator (e.g., the original configuration file for "inserter" is insert_nums: 1, insert_ratio: 0, insert_time_sort: maximum, percent_no_augment: 0, ti_insert_n_times: 1). In a zero-shot scenario, the model explores optimization strategies based on dataset characteristics and the meaning of the operators, with a probability of converging to a local optimum—i.e., finding a suboptimal combination and deeming the result sufficient, thus stopping the exploration of new combinations and selecting the best operator configuration based on this combination. We compared the llama3-70b used in the paper with other reasoning models, and the results are shown in Table 11 (S2TUL, FSK-London, memory-length=1). We found that models with stronger reasoning capabilities attempt more operator combinations and have a higher probability of finding better combinations in fewer steps. For instance, DeepSeek-v3 outperforms LLaMA-3-70B in step 4. Under the same operator combination and the same number of steps, models with stronger reasoning capabilities yield better optimization results. For instance, DeepSeek-v3 outperforms LLaMA-3-70B in steps 3, 5, 6, 7, 9, 10, 11, and 13.

**Improvement solution**: Implement reflection similar to contrastive learning between steps, such as further adjusting the parameters of each operator based on effective combinations, to avoid exploring ineffective combinations as much as possible. The improved results are shown in Table 12 (S2TUL, FSK-London, memory-length=1).

**The optimization effect diminishes as the memory length increases**: memory length refers to the number of action proposals generated in each reasoning step. Increasing it can improve exploration efficiency but also raises the probability of falling into a local optimum. We compared the performance of llama3-70b with that of reasoning models, and the results are shown in Table 9 (S2TUL, FSK-London, step=5). Both models achieved relatively good results at memory_length=2, but as the length increased, they faced the issue of converging to suboptimal solutions. However, models with stronger reasoning capabilities exhibited a stronger tendency to explore other combinations, thus having a higher probability of finding better combinations and escaping the "optimization trap" .

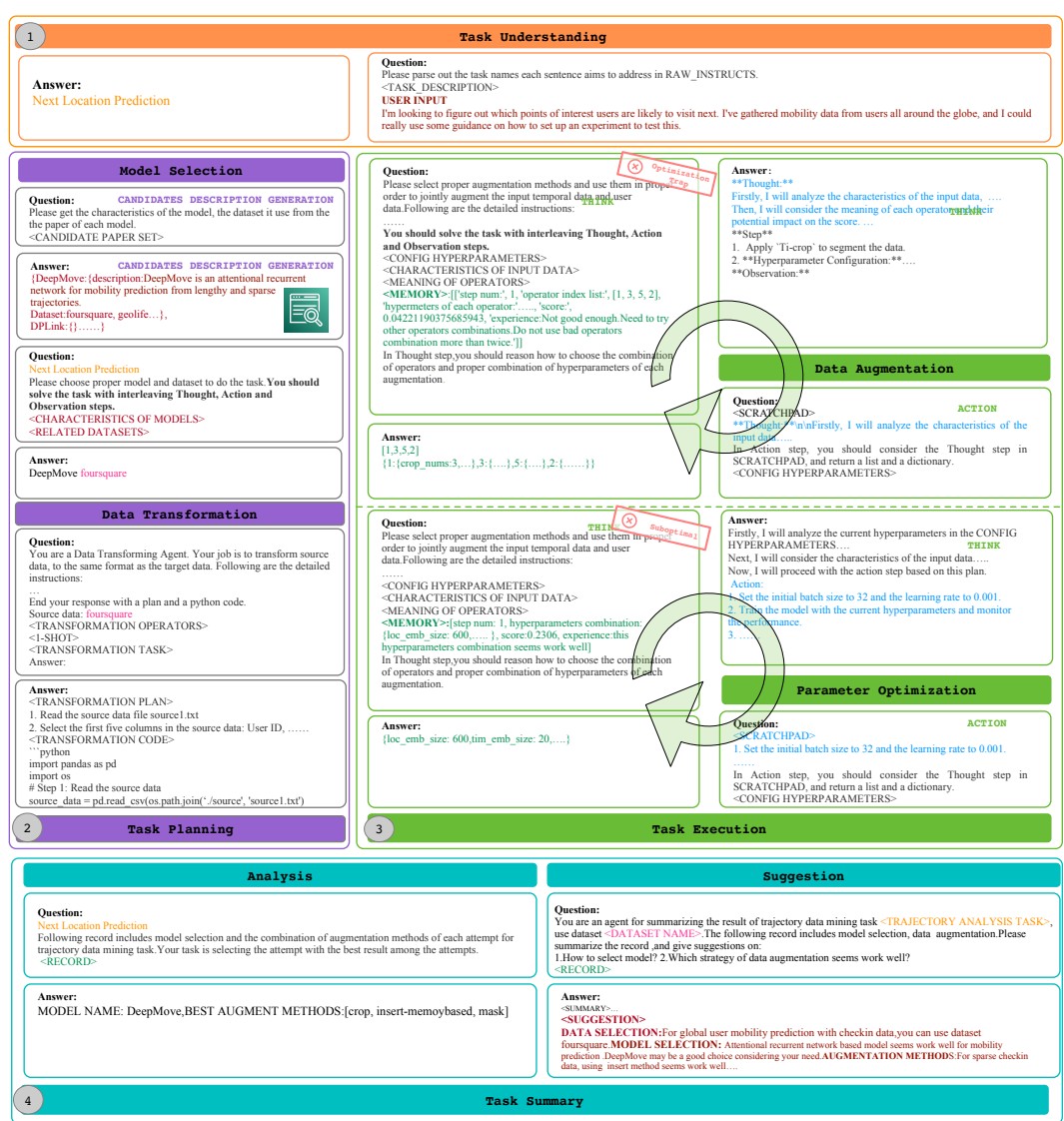

Figure 5: A representative example case of *TrajAgent*.

Table 8: Token consumption and time cost for trajectory processing with different models (traj = 200). Input and output token counts are reported for two inference settings: step=1 and step=5 (with memory_length=1).

| Model | token(step=1) | | token(step=5) | | time cost |
|---|---|---|---|---|---|
| | input | output | input | output | |
| LLM-ZS | 37,327 | 90,180 | 194,358 | 282,270 | 3h27min |
| DutyTTE | 1,108 | 284 | 7,718 | 1,823 | 1h17min |

**Improvement solution**: Periodically optimize and update the memory organization, discard poor combinations, retain good ones, and guide the model to explore new combinations during the reflection phase. The improved results are shown in Table 11 (S2TUL, FSK-London, step=5).

Table 9: Step-wise ACC@5 performance and Operator Combination Order(OCO) traces for multiple large reasoning models.

| step | 0 | 1 | 2 | 3 | 4 | 5 | 6 | 7 | 8 | 9 | 10 | 11 | 12 | 13 | 14 | 15 | 16 | 17 |
|---|---|---|---|---|---|---|---|---|---|---|---|---|---|---|---|---|---|---|
| **Llama3-70b** | | | | | | | | | | | | | | | | | | |
| ACC@5 | 56.28 | 56.28 | 59.10 | 68.18 | 68.18 | 68.39 | 68.18 | 69.04 | 69.04 | 68.39 | 69.04 | 69.04 | 87.01 | 69.05 | 68.40 | 68.19 | 83.12 | 88.96 |
| OCO | [] | [] | [1,3,9] | [1,3,9] | [1,3,9] | [1,3,9] | [1,3,9] | [1,3,9] | [1,3,9] | [1,3,9] | [1,3,9] | [1,3,9] | [2,9,10] | [1,3,9] | [1,3,9] | [1,3,9] | [2,9,10] | [2,9,10] |
| **DeepSeek-v3** | | | | | | | | | | | | | | | | | | |
| ACC@5 | 56.28 | 79.22 | 59.52 | 61.26 | 81.17 | 68.83 | 69.26 | 69.05 | 83.98 | 70.35 | 69.05 | 69.26 | 71.00 | 61.90 | 60.82 | 68.83 | 71.00 | 66.88 |
| OCO | [] | [2,9,10] | [1,3,9] | [1,3,9] | [2,9,10] | [1,3,9] | [1,3,9] | [1,3,9] | [2,9,10] | [1,3,9] | [1,3,9] | [1,3,9] | [1,3,9] | [1,3,9] | [1,4,9] | [1,3,9] | [1,3,9] | [1,3,5,9] |
| **Gemini-2.0-flash-001** | | | | | | | | | | | | | | | | | | |
| ACC@5 | 56.28 | 73.16 | 61.69 | 70.35 | 68.83 | 82.25 | 69.05 | 68.40 | 87.01 | 65.58 | 79.87 | 83.12 | 71.65 | 84.63 | 82.47 | 71.43 | 69.05 | 69.05 |
| OCO | [] | [2,9,10] | [1,3,9] | [1,3,9] | [1,3,9] | [3,9,10] | [1,3,9] | [1,3,9] | [2,9,10] | [1,3,9] | [2,9,10] | [2,9,10] | [1,3,9] | [2,9,10] | [2,9,10] | [1,3,9] | [1,3,9] | [1,3,9] |

Table 10: Performance (ACC@5) and Operator Combination Order(OCO) traces under different memory lengths for Llama3-70b and DeepSeek-v3.

| memory-length | 0 | 1 | 2 | 3 | 4 | 5 | 6 | 7 |
|---|---|---|---|---|---|---|---|---|
| **Llama3-70b** | | | | | | | | |
| ACC@5 | 56.28 | 68.39 | 80.95 | 79.00 | 55.19 | 83.33 | 57.79 | 62.55 |
| OCO | [] | [1,3,9] | [2,9,10] | [2,9,10] | [1,2,9] | [2,9,10] | [1,4,9] | [2,6,10] |
| **DeepSeek-v3** | | | | | | | | |
| ACC@5 | 56.28 | 79.22 | 59.09 | 79.65 | 61.47 | 80.74 | 59.52 | 81.60 |
| OCO | [] | [2,9,10] | [1,3,9] | [3,6,9] | [2,5,9] | [2,9,10] | [1,4,9] | [2,9,10] |

Table 11: Raw step-wise performance and Operator Combination Order(OCO) traces for llama3-70b before and after improvement.

| step | 0 | 1 | 2 | 3 | 4 | 5 | 6 | 7 | 8 | 9 | 10 | 11 |
|---|---|---|---|---|---|---|---|---|---|---|---|---|
| **Pre-improvement** | | | | | | | | | | | | |
| ACC@5 | 56.28 | 56.28 | 59.10 | 68.18 | 68.18 | 68.39 | 68.18 | 69.04 | 69.04 | 68.39 | 69.04 | 69.04 |
| OCO | [] | [] | [1,3,9] | [1,3,9] | [1,3,9] | [1,3,9] | [1,3,9] | [1,3,9] | [1,3,9] | [1,3,9] | [1,3,9] | [1,3,9] |
| **Post-improvement** | | | | | | | | | | | | |
| ACC@5 | 56.28 | 59.09 | 61.03 | 68.18 | 69.04 | 69.04 | 60.38 | 69.04 | 82.90 | 80.08 | 87.01 | 83.12 |
| OCO | [] | [1,3,9] | [1,3,9] | [1,3,9] | [1,3,9] | [1,3,9] | [1,3,9] | [1,3,9] | [2,9,10] | [2,9,10] | [2,9,10] | [2,9,10] |

Table 12: Performance and tool invocation Operator Combination Order(OCO) traces under different memory lengths before and after improvement.

| memory-length | 0 | 1 | 2 | 3 | 4 | 5 | 6 | 7 |
|---|---|---|---|---|---|---|---|---|
| **Pre-improvement** | | | | | | | | |
| ACC@5 | 56.28 | 68.39 | 80.95 | 79.00 | 55.19 | 83.33 | 57.79 | 62.55 |
| OCO | [] | [1,3,9] | [2,9,10] | [2,9,10] | [1,2,9] | [2,9,10] | [1,4,9] | [2,6,10] |
| **Post-improvement** | | | | | | | | |
| ACC@5 | 56.28 | 68.39 | 69.05 | 80.95 | 81.60 | 84.63 | 80.52 | 85.73 |
| OCO | [] | [1,3,9] | [1,3,9] | [2,9,10] | [2,9,10] | [2,9,10] | [2,9,10] | [3,6,9] |

Table 13: Performance comparison of TrajAgent with and without utilizing score feedback in memory across optimization steps.

| Step | 0 | 1 | 2 | 3 | 4 | 5 | 6 | 7 | 8 | 9 | 10 | 11 | 12 |
|---|---|---|---|---|---|---|---|---|---|---|---|---|---|
| **With score in memory** | | | | | | | | | | | | | |
| ACC@5 (%) | 56.28 | 56.28 | 59.10 | 68.18 | 68.18 | 68.39 | 68.18 | 69.04 | 69.04 | 68.39 | 69.04 | 69.04 | 87.01 |
| Operators | [] | [] | [1,3,9] | [1,3,9] | [1,3,9] | [1,3,9] | [1,3,9] | [1,3,9] | [1,3,9] | [1,3,9] | [1,3,9] | [1,3,9] | [2,9,10] |
| **Without score in memory** | | | | | | | | | | | | | |
| ACC@5 (%) | 56.28 | 45.23 | 78.14 | 47.40 | 48.05 | 57.36 | 47.61 | 58.22 | 58.23 | 58.87 | 58.87 | 46.32 | 63.85 |
| Operators | [] | [1,4,9] | [2,9,10] | [1,4,9] | [1,4,9] | [1,4,9] | [1,4,9] | [1,4,9] | [1,4,9] | [1,4,9] | [1,4,9] | [1,4,9] | [2,6,10] |

## A.7 Prompt Examples

**Parameter Optimization**

**User**

Please select proper combination of hyperparameters of model in CONFIG HYPERPARAME-TERS.Adjust the selection, the combination of hyperparameters based on the main function of hyperparameters, the characteristics of the input data, the tuning principles, and memory to get a high score. You should solve the task with interleaving Thought, Action and Observation steps.

<CHARACTERISTICS OF INPUT DATA>

The input temporal data contains a time dictionary(key is the user ID,the value is a list containing all time points when the user is active in chronological order) , the input user data contains a user dictionary(key is the user ID,the value is a list containing all items that the user interacts with in chronological order).

<CONFIG HYPERPARAMETERS>

```
{loc_emb_size: 500, tim_emb_size: 10, ...}
```

<TUNING PRINCIPLES>

1.Start with a small batch size (32-64) and a small learning rate (0.001-0.01): This will help prevent overshooting and overfitting.

2.Increase batch size and learning rate: If the model is not overfitting, increasing the batch size and learning rate can help improve convergence.

3.Add dropout (0.2-0.5) to prevent overfitting: If the model is overfitting, adding dropout can help regularize the model.

4.Increase embedding size: If the model is not capturing enough information, increasing the embedding size can help improve representational power.

5.Decrease learning rate and increase batch size: If the model is not converging, decreasing the learning rate and increasing the batch size can help improve stability.

<MEMORY>:

<SCRATCHPAD>:

In Thought step,you should reason how to choose the combination of hyperparameters to get a higher score.Please consider following aspects:

1.Observe the hyperparameters with high scores in MEMORY, to determine the optimal hyper-parameters.

2.Use a grid search: Perform a grid search over a range of hyperparameters with high scores to find the optimal combination with higher scores.

3.Increase batch size and learning rate, and use dropout to avoid overfitting.

4. Stop or reverse the adjusting trend if the score is decreasing. According to above aspects,please first learn experiences from MEMORY, then make plan for the action step.Please use the sentence structure 'Firstly... Then... Lastly'.Let's think step by step.

Thought:

**Assistant**

Firstly, I will analyze the current hyperparameters in the CONFIG HYPERPARAMETERS. ...

Then, I will consider the hidden size, which is currently set at 500. ...

Next, the learning rate is currently at 0.001, which is a good starting point. ...

Lastly, I will monitor the model's performance closely....

Now, I will proceed with the action step to adjust the hyperparameters based on this reasoning.

Action:

1. Increase the tim_emb_size from 10 to 20 to capture more temporal information.

2. Keep the loc_size at 500 for now, as it seems adequate.

3. Increase the hidden_size from 500 to 600 to enhance the model's capacity.

4. Increase the dropout_p from 0.3 to 0.4 to further prevent overfitting.

5. Set the learning_rate to 0.005 to improve convergence speed.

6. Start with a batch size of 32 and monitor the performance closely.

Observation:

After implementing these changes, I will evaluate the model's performance on a validation set. ...

**User**

In Action step, you should consider the Thought step in SCRATCHPAD, and give a dict:{hypermeter name:hypermeter value}.The hypermeter name should be the same with the raw config hypermeter names in CONFIG HYPERPARAMETERS,and hypermeter values should be the same type as the hypermeter values in CONFIG HYPERPARAMETERS.Please do not add any comments to each value.

Action:

**Assistant**

{'loc_emb_size': 500,...}

**Task Understanding**

**User**

The description of each task is in TASK_DESCRIPTION.

Please parse out the task name the sentence aims to address in RAW_INSTRUCTS.

1. The task name should match the key in the DESCRIPTION.

2. Please only output the task name.Do not output other contents.

<TASK_DESCRIPTION>

{Next_Location_Prediction: ...,Trajectory_User_Linkage:...}

<RAW_INSTRUCT>

I want to identify the trajectory...

**Assistant**

Trajectory_Completion

**Data Augmentation**

**User**

<TASK>

Please:

1.select proper augmentation methods and use them in proper order to jointly augment the input temporal data and user data.Please adjust the selection and combination sequence of operators based on MEANING OF OPERATORS, CHARACTERISTICS OF INPUT DATA, and MEMORY to get a high score.

2.select proper combination of hyperparameters of each augmentation method in CONFIG HYPERPARAMETERS.Adjust the selection, the combination of hyperparameters based on the main function of hyperparameters, the characteristics of the input data, and memory to get a high score.

You should solve the task with interleaving Thought, Action and Observation steps.

<CONFIG HYPERPARAMETERS>

1.Ti-crop: crop_nums:(int) the size of cropping. default is 2.

crop_ratio:(float) the ratio of cropping. default is 0.

crop_n_times:(int) the number of cropping for each sequence for default setting. default is 2.

crop_time_sort:(str, choice in [maximum, minimum]) choose the candidate subsequence in a descending/ascending order according to its time interval variance. default is 'minimum'. ...

2.Ti-insert_unvisited: ...

<MEANING OF OPERATORS>

1.Ti-crop: For each user, filter out sessions with fewer trajectory points than ti_threshold + ti_crop_n_times. In each session, randomly extract several segments (ti_crop_n_times) of equal length (ti_threshold). Sample data segments according to the rules.

2.Ti-insert_unvisited:...

...

<MEMORY>:

<SCRATCHPAD>:

In Thought step,you should reason how to choose the combination of operators and proper combination of hyperparameters of each augmentation method to get a higher score.Please consider following aspects:

1.The meaning of each operator,and the impact of adding or removing operators at a specific location on the score.

2.Common characteristics of operator index lists with higher scores in MEMORY.Avoid using the same index list as MEMORY with scores lower than 0.1811.

3.The meaning of hyperparameters of each operator you select, and how to adjust the hyperparameters based on the characteristics of input data and hyperparameters with high scores in MEMORY.

4.Use a grid search: Perform a grid search over a range of hyperparameters with high scores to find the optimal combination with higher scores.

7. Stop or reverse the adjusting trend if the score is decreasing.

According to above aspects,please first learn experiences from MEMORY, then make plan for the action step.Please use the sentence structure 'Firstly... Then... Lastly'.Let's think step by step.

Thought:

**assistant**

**Thought:**

Firstly, I will analyze the characteristics of the input data, which consists of user activity timestamps and the corresponding items they interacted with. This suggests that temporal relationships and user preferences are crucial for effective augmentation. The augmentation methods should enhance these aspects without distorting the underlying patterns.

Then, I will consider the meaning of each operator and their potential impact on the score. For instance, using 'Ti-crop' can help in segmenting the data, which is essential for understanding user behavior over time. ...

Lastly, I will select hyperparameters for each operator based on their functions and the characteristics of the input data.

**Action:**

1. **Select Augmentation Methods:** - **Ti-crop**:

- 'crop_nums': 3

- 'crop_ratio': 0.1 (to allow for a small portion of the data to be cropped)

- 'crop_n_times': 3

...

2. **Hyperparameter Configuration:**

- For 'Ti-crop', I will keep 'crop_n_times' at 3 to ensure multiple segments are created....

3. **Evaluate and Adjust:**

- After implementing the above configurations, I will evaluate the score...

**Observation:**

I will monitor the score after each adjustment to ensure that the changes lead to an improvement. If the score decreases, I will revert to the previous settings.

**User**

<SCRATCHPAD>

**Action:**

1. **Select Augmentation Methods:**

- **Ti-crop**: ...

In Action step, you should consider the Thought step in SCRATCHPAD, and return a list and a dictionary.

The list should contain the indices of augmentation methods in {1: 'Ti-crop', 2: 'Ti-insert_unvisited...}.For example, if you want to first use Ti-crop, secondly use Ti-insert_random, thirdly use Ti-mask, then the list should be '[1,4,8]'.

The dict:....The hypermeter name should be the same with the corresponding operator hypermeter names in CONFIG HYPERPARAMETERS,and hypermeter values should be the same type as the corresponding operator hypermeter values in CONFIG HYPERPARAMETERS.

Please directly output the list and dictionary.For example:...

Action:

**Assistant**

[1, 3, 5]

{{crop_nums:3,crop_ratio:0,....},3:{...},5:{...}}

