# OpenReview forum: "TrajAgent: An LLM-Agent Framework for Trajectory Modeling via Large-and-Small Model Collaboration"
_NeurIPS.cc/2025/Conference — NeurIPS 2025 poster_

### Official Review · Reviewer_PauW · 2025-06-29

**Clarity:** 2
**Significance:** 4
**Originality:** 3
**Rating:** 4
**Confidence:** 4

**Summary:**

This paper proposes an LLM-Agent for trajectory modeling for various practical applications, including next location prediction, travel time estimation, trajectory generation, and many other relevant tasks. The proposed framework incorporates deep learning models and LLM-based models, utilizing collaborative learning to address to effectively enhance the performance.  The proposed unified running environment can integrate various models and process diverse trajectory data. Experimental results on various trajectory tasks demonstrate that the proposed unified framework achieves superior performance compared to other recent state-of-the-art methods.

**Questions:**

Tiny weakness:

1.	T in Eq. (1) indicates task?

2.	More citations should be provided to support the assumption “Following the common practice[38]”.

3.	Typos and grammatical errors. For example,  “agentic framework [44, 41, 50] are” in Line 360 of the main manuscript.
4.	Some recent related works are missing:

[1] Zhu, Y., Yu, J. J., Zhao, X., Liu, Q., Ye, Y., Chen, W., ... & Liang, Y. (2024, August). Controltraj: Controllable trajectory generation with topology-constrained diffusion model. In Proceedings of the 30th ACM SIGKDD Conference on Knowledge Discovery and Data Mining (pp. 4676-4687).
[2] Mao, X., Lin, Y., Guo, S., Chen, Y., Xian, X., Wen, H., ... & Wan, H. (2025, April). DutyTTE: Deciphering Uncertainty in Origin-Destination Travel Time Estimation. In Proceedings of the AAAI Conference on Artificial Intelligence (Vol. 39, No. 12, pp. 12390-12398).

The paper presents an interesting framework that incorporates LLM-based models and deep learning models, achieving state-of-the-art performance on various trajectory tasks. The manuscript is well-organized and clearly described. Nevertheless, integrating outdated models weakens its effectiveness and superiority. The authors should provide more technical details, ablation experiments, and deeper analyses to support their assumptions. Additionally, the authors should thoroughly review and revise the manuscript to enhance the rigor in writing and architecture diagrams. Therefore, I vote for borderline reject in this review round. I would like to raise the rating if the authors can address my concerns.

**Ethical Concerns:**

["NO or VERY MINOR ethics concerns only"]

**Final Justification:**

The responses have addressed my concerns, therefore, I would like to raise my rating.

**Limitations:**

The authors have discussed the limitations in the supplemental material.

**Paper Formatting Concerns:**

None.

**Quality:**

3

**Strengths And Weaknesses:**

Paper Strengths:

The main insight of this work is the unified framework which can integrate various LLM-based or deep learning models to address various trajectory tasks. The proposed framework utilizes collaborative learning to model collaborations between LLM-based models and deep learning models, enhancing the performance of various practical applications. The developed unified running environment is interesting which can integrate diverse state-of-the-art models.

Paper Weaknesses:

1.	My first concern is the technical details of each core module and architecture diagrams. The authors introduce the proposed collaborative learning in Sec. 2.4 Collaborative Learning Schema, without sufficient technical details and the underlying motivations. Some ablation experiments should be conducted to examine the effectiveness of collaborative learning schema since this module is crucial for combining LLM-based models and deep learning models. Additionally, the architecture diagrams seem confusing. In the right part of Figure 1, the connection lines for T2 and T4 have no arrows. The dotted line in Figure 1 is confusing. In the left part of Figure 2 (UniEnv), the descriptions or definitions should be included for colored rectangles below Trajectory Prediction. The authors need to modify the figures to improve the rigor.

2.	My second concern is the effectiveness of the proposed framework. Some of the integrated methods are outdated to effectively demonstrate the capabilities of the proposed TrajAgent. The authors should integrate more recent state-of-the-art methods to evaluate the performance improvement. For example, integrating DutyTTE [Mao et al., AAAI 2025] for travel time estimation and ControlTraj [Zhu et al., ACM SIGKDD 2024] for trajectory generation.

3.	I have some concerns on the literature review. Sec. 4 Related Work is located at the last page of the manuscript, is there any specific meaning in organizing the manuscript? The summarized papers in related work should be updated. Including more recent researches and analyses may be better. Additionally, the authors should thoroughly check the references since some of mentioned methods have been accepted by conferences or journals. For example, MetaGPT [18] has been accepted by ICLR 2024, TrajectoryNet [20] was published in Annual International Conference on Computer Science and Software Engineering, and UrbanLLM [21] has been published in Findings of the Association for Computational Linguistics: EMNLP 2024.

4.	Some concerns on experimental results. Firstly, in Table 1 of the manuscript, the authors show the performance improvements across various trajectory tasks and examine the effectivenss of core optimization methods (e.g., data augmentation). However, the performances of some operations (e.g., +DA and +PO for LLM-ZS) are missing. Secondly, in Figure 4(a-b), the authors conduct the ablation experiment on thought steps and memory size. The performance is compard in terms of Accuracy@5 in next location prediction task. With the increase of thought steps and memory size, the performance of proposed framework is significantly improved. However, with excessive thought steps (e.g., thought step = 20) and larger memory size (e.g., memory length =15), the performance decreases without any analyses. The authors need to include more deeper analyses. Thirdly, in A.5 Case Study of the supplemental material, the limitations are discussed on two specific challenging issues. Including some experiments to explore possible reasons and future work would make the manuscript more persuasive.

---

> ### Author Rebuttal · Authors · 2025-07-31
>
> ## For W1
> **Ablation experiments** During the task execution phase, our goal is to leverage agentic framework to drive small models in completing various trajectory modeling tasks that LLMs are not good at, while further optimizing the performance of small models. We compared the effects of independent execution and TrajAgent joint execution, with the experimental results presented in Table 1. In the table, "-DL" represents our implementation method designed with reference to other large model approaches, "-LLM" denotes the small model, and "Joint" refers to the result of large models driving small models with optimization.  MAE=100% and Acc@5=0 values in the table suggest that no effective prompt-based LLM approaches were found for these tasks.
>
> **Figure** Thank you for your suggestions regarding the figures. We apologize for omitting some arrows in Figure 1. The direction of the arrows aligns with the flow of data/information, originating from the raw data (Data module) and input into the TrajAgent framework (Framework module). The large model automatically performs data preprocessing, small model selection, and small model deployment training based on the input data type and task description, then dynamically selects the optimal optimization strategy based on the training results, ultimately producing an experimental solution applicable to various downstream tasks (Task module). The left side of the dashed line represents the traditional single-model paradigm, where a single model is designed for a specific task and is only suitable for particular datasets. The right side of the dashed line represents our solution, which constructs a unified trajectory modeling framework adaptable to multiple tasks and datasets. In Figure 2, the colored blocks represent the currently integrated models for each type of task. Since the models and datasets will be further expanded in the future, the names of each model are not labeled.
> | Task(metrics, dataset)          | Trajectory Prediction<br>(Acc@5, FSQ-NYC) | Trajectory Recovery<br>(MAE, Porto) | Trajectory Classification<br>(Acc@5, FSQ-NYC) | Trajectory Generation<br>(JSD-MEAN, FSQ) | Trajectory Representation<br>(Acc@5, FSQ-NYC) |
> |-|-|-|-|-|-|
> | remove DL model | 0.3950| 100% | 0.2034 | 0.3200| 0.00 |
> | remove LLM | 0.3422| 42.71% | 0.5755  | 0.4557| 0.3100|
> | TrajAgent| 0.4018 | 27.78% | 0.7802  | 0.3200 | 0.3472 |
>
> Table 1. Ablation experiments
> ## For W2
> Thank you for your suggestion. We have integrated the **DutyTTE** model you proposed. For the MoEUQ method, the optimization results are shown in Table 2.  TrajAgent autonomously selects key parameters and configurations.
>
> | Model | GraphMM (Acc@5) | LLM-ZS (Acc@5) | DutyTTE (MAE) |
> |-|-|-|-|
> | Dataset | Tencent | FSQ-TKY | Tencent |
> | Origin | 0.2014 | 0.3110 | 190.82 |
> | +DA | 0.3258 | -  | - |
> | +PO | 0.2427 | 0.3302| 179.01 |
> | +Prompt | - | 0.3225 | - |
> | Joint | 0.3422 | 0.3350 | 179.01 |
>
> Table 2. Expanding models and optimization methods
> ## For W3
> Thank you for your reminder and suggestions. We apologize for not updating the information of the cited articles in a timely manner, which was an oversight during the resubmission process. We have updated this information in the latest version of the paper, along with the recent work mentioned in the "tiny weakness" section.
>
> ## For W4
> Thank you for your question!
> - **Extended Models**
>     For the LLM-based method (e.g., LLM-ZS), since it uses a pre-trained model and the trajectory data does not participate in training, no data augmentation (DA) method is provided. We allowed TrajAgent to automatically optimize the key parameters mentioned in the original paper, and the results are shown in Table 2. For other small models using map/GPS trajectory data(e.g., GraphMM), we adopt the rule-based data augmentation approach from DeepMM, which first down-samples the raw data, then designs various noise-injection rules, and finally combines the processed data with the original dataset to achieve data augmentation. We integrated it into the TrajAgent framework, using agent to choose noise-injection strategies. The experimental results are shown in Table 2 (base_model=llama3-70b, step=5).
>
> - **The optimization effect diminishes as the step increases**
>
>     While selecting combinations of operators, the model further optimizes the configuration of each operator (e.g., the original configuration file for "inserter" is {insert_nums: 1, insert_ratio: 0, insert_time_sort: maximum, percent_no_augment: 0, ti_insert_n_times: 1}). In a zero-shot scenario, the model explores optimization strategies based on dataset characteristics and the meaning of the operators, with a probability of converging to a local optimum—i.e., finding a suboptimal combination and deeming the result sufficient, thus stopping the exploration of new combinations and selecting the best operator configuration based on this combination. We compared the llama3-70b used in the paper with other reasoning models, and the results are shown in Table 3 (S2TUL, FSK-London, memory-length=1). We found that models with stronger reasoning capabilities attempt more operator combinations and have a higher probability of finding better combinations in fewer steps. For instance, DeepSeek-v3 outperforms LLaMA-3-70B in step 4. Under the same operator combination and the same number of steps, models with stronger reasoning capabilities yield better optimization results. For instance, DeepSeek-v3 outperforms LLaMA-3-70B in steps 3, 5, 6, 7, 9, 10, 11, and 13.
>
>     - **Improvement solution**
>
>     Implement reflection similar to contrastive learning between steps, such as further adjusting the parameters of each operator based on effective combinations, to avoid exploring ineffective combinations as much as possible. The improved results are shown in Table 4 (S2TUL, FSK-London, memory-length=1).
>
> - **The optimization effect diminishes as the memory_length increases**
>
>     Memory_length refers to the number of action proposals generated in each reasoning step. Increasing it can improve exploration efficiency but also raises the probability of falling into a local optimum. We compared the performance of llama3-70b with that of reasoning models, and the results are shown in Table 5 (S2TUL, FSK-London, step=5). Both models achieved relatively good results at memory_length=2, but as the length increased, they faced the issue of converging to suboptimal solutions. However, models with stronger reasoning capabilities exhibited a stronger tendency to explore other combinations, thus having a higher probability of finding better combinations and escaping the "optimization trap" .
>
>     - **Improvement solution**
>
>      Periodically optimize and update the memory organization, discard poor combinations, retain good ones, and guide the model to explore new combinations during the reflection phase. The improved results are shown in Table 6 (S2TUL, FSK-London, step=5).
> | base-model| step | 0| 1 | 2 | 3 | 4 | 5| 6| 7| 8 | 9| 10| 11| 12| 13| 14| 15| 16| 17|
> |-|-|-|-|-|-|-|-|-|-|-|-|-|-|-|-|-|-|-|-|
> | llama3-70b | ACC@5| 56.28 | 56.28 | 59.10 | 68.18 | 68.18 | 68.39 | 68.18 | 69.04 | 69.04 | 68.39 | 69.04 | 69.04 | 87.01 | 69.05 | 68.40 | 68.19 | 83.12 | 88.96 |
> | deepseek-v3 | ACC@5| 56.28 | 79.22 | 59.52 | 61.26 | 81.17 | 68.83 | 69.26 | 69.05 | 83.98 | 70.35 | 69.05 | 69.26 | 71.00 | 61.90 | 60.82 | 68.83 | 71.00 | 66.88 |
> | gemini-2.0-flash-001  | ACC@5| 56.28 | 73.16 | 61.69 | 70.35 | 68.83 | 82.25 | 69.05 | 68.40 | 87.01 | 65.58 | 79.87 | 83.12 | 71.65 | 84.63 | 82.47 | 71.43 | 69.05 | 69.05 |
> Table 3. Comparative analysis of TrajAgent performance across optimization steps
>
> | Base-Model  | Version | Metric | Step 0 | Step 1 | Step 2 | Step 3 | Step 4 | Step 5 | Step 6 | Step 7 | Step 8 | Step 9 | Step 10 | Step 11 |
> |-|-|-|-|-|-|-|-|-|-|-|-|-|-|-|
> | llama3-70b   | Pre-improvement  | ACC@5  | 56.28  | 56.28  | 59.10  | 68.18  | 68.18  | 68.39  | 68.18  | 69.04  | 69.04  | 68.39  | 69.04 | 69.04|
> | | Post-improvement  | ACC@5  | 56.28  | 59.09  | 61.03  | 68.18  | 69.04  | 69.04  | 60.38  | 69.04  | 82.90  | 80.08  | 87.01 | 83.12|
> Table 4. Improvement
>
> | Base-Model   | Metric   | Memory Length 0 | Memory Length 1 | Memory Length 2 | Memory Length 3 | Memory Length 4 | Memory Length 5 | Memory Length 6 | Memory Length 7 |
> |-|-|-|-|-|-|-|-|-|-|
> | llama3-70b   | ACC@5| 56.28 | 68.39 | 80.95| 79.00| 55.19| 83.33 | 57.79 | 62.55 |
> | deepseek-v3  | ACC@5| 56.28 | 79.22| 59.09| 79.65| 61.47 | 80.74| 59.52 | 81.60 |
> Table 5. Comparative analysis of TrajAgent performance across memory lengths
>
> | Base-Model   | Version | Metric   | Memory Length 0 | Memory Length 1 | Memory Length 2 | Memory Length 3 | Memory Length 4 | Memory Length 5 | Memory Length 6 | Memory Length 7 |
> |-|-|-|-|-|-|-|-|-|-|-|
> | llama3-70b   | Pre-improvement  | ACC@5 | 56.28| 68.39| 80.95  | 79.00 | 55.19 | 83.33 | 57.79| 62.55 |
> | | Post-improvement  | ACC@5| 56.28 | 68.39 | 69.05| 80.95| 81.60 | 84.63 | 80.52 | 85.73 |
> Table 6. Improvement

---

> > ### Comment · Reviewer_PauW · 2025-08-05
> >
> > Thanks for the responses, which have addressed most of my concerns. The additional experiments further demonstrate the effectiveness of the proposed method. I have one more question, how integrated models selected? In the supplementary material, the authors provided the adopted models without any descriptions explaining the selection criteria.

---

> > > ### Author Response · Authors · 2025-08-05
> > > **Response with Gratitude: Clarification on Model Selection Mechanism**
> > >
> > > We sincerely appreciate your feedback and apologize for not providing sufficient details about our model selection mechanism in the original manuscript. Our model selection module operates through a two-phase process:
> > >
> > > - **Task-Dataset Alignment Phase**
> > >
> > >     Most trajectory modeling models are designed for specific tasks/datasets. Model selection module implement following procedural steps:
> > >
> > >     - Employ an commercial LLM specialized for academic literature analysis to extract key information including problem definition, model architecture, and dataset specifications from each model's original publication.
> > >     - Struct information into a dictionary: {model name: model description}.
> > >     - Perform matching between the task identification module's outputs and the dictionary to select the most appropriate candidate models.
> > >
> > > - **Performance Optimization Phase**
> > >
> > >     Since actual model performance is highly dependent on trajectory dataset characteristics and parameter configurations, relying solely on citation counts for model selection proves inadequate. We implement an empirical optimization process:
> > >
> > >     - After initial selection, TrajAgent executes the complete pipeline while storing optimal configurations and corresponding performance metrics.
> > >     - The model selection module iteratively refines selection through multiple execution cycles.
> > >     - The analysis module subsequently examines these execution records to determine the  model that best satisfies user requirements and its associated optimal configuration.
> > >
> > > In future work, we plan to investigate early-stopping mechanisms to reduce the computational overhead of the trial-and-error optimization process, while maintaining the robustness of model selection. We sincerely appreciate your valuable questions and insights once again.

---

> > > > ### Comment · Reviewer_PauW · 2025-08-06
> > > >
> > > > Thanks for the responses. I have no further questions. I suggest that the authors could include these descriptions in the supplementary materials.

---

> > > > > ### Author Response · Authors · 2025-08-06
> > > > >
> > > > > Thank you for your thorough review and valuable insights on our work. We're glad our responses have addressed all questions, as reflected in your kind confirmation that there are no further questions. We recall your earlier indication that satisfactory responses could warrant a score adjustment, and would be grateful if you might now consider updating the evaluation accordingly. As suggested, we've incorporated these clarifications in the supplementary materials, and remain available for any additional discussion needed.

---

> ### Author Response · Authors · 2025-08-03
> **Supplementary compilation of reply contents and request for further discussion**
>
> We sincerely appreciate your valuable questions. For your convenience, we have summarized the questions that were previously simplified-away in the main response or contained in other reviewers' responses impacted by length constraints. We would be grateful if you could kindly confirm whether it satisfactorily resolves your concerns. Should the explanations meet your expectations, we would deeply appreciate your consideration for score improvement. If any uncertainties remain, we are fully prepared to provide additional clarifications through further discussion to eliminate any doubts and collectively enhance the paper's quality.
> ## Complete versions of Tables 4 to 6
> The "detailed" column specifies the selection sequence of data augmentation operators, where each numeric code represents a specific operator. For example, [1, 2, 9] indicates that the data should undergo crop, insert_unvisited, and reorder operations sequentially. Additionally, TrajAgent needs to configure detailed parameters for each operator, such as:
> ```python
> crop: {crop_n_times: 2, crop_nums: 2, crop_ratio: 0, crop_time_sort: minimum, ti_crop_n_times: 3}
> ```
>
> | Base-Model  | Version | Metric | Step 0 | Step 1 | Step 2 | Step 3 | Step 4 | Step 5 | Step 6 | Step 7 | Step 8 | Step 9 | Step 10 | Step 11 |
> |-|-|-|-|-|-|-|-|-|-|-|-|-|-|-|
> | llama3-70b   | Pre-improvement  | ACC@5  | 56.28  | 56.28  | 59.10  | 68.18  | 68.18  | 68.39  | 68.18  | 69.04  | 69.04  | 68.39  | 69.04 | 69.04|
> ||| detailed | []| []| [1,3,9] | [1,3,9] | [1,3,9] | [1,3,9] | [1,3,9] | [1,3,9] | [1,3,9] | [1,3,9] | [1,3,9]  | [1,3,9]|
> | | Post-improvement  | ACC@5  | 56.28  | 59.09  | 61.03  | 68.18  | 69.04  | 69.04  | 60.38  | 69.04  | 82.90  | 80.08  | 87.01 | 83.12|
> | || detailed | [] | [1,3,9] | [1,3,9] | [1,3,9] | [1,3,9] | [1,3,9] | [1,3,9] | [2,9,10] | [2,9,10] | [2,9,10] | [2,9,10]| [2,9,10]|
> Table 4. Improvement
>
> | Base-Model   | Metric   | Memory Length 0 | Memory Length 1 | Memory Length 2 | Memory Length 3 | Memory Length 4 | Memory Length 5 | Memory Length 6 | Memory Length 7 |
> |-|-|-|-|-|-|-|-|-|-|
> | llama3-70b   | ACC@5| 56.28 | 68.39 | 80.95| 79.00| 55.19| 83.33 | 57.79 | 62.55 |
> | | detailed | []| [1,3,9] | [2,9,10]| [2,9,10]| [1,2,9]| [2,9,10]| [1,4,9] | [2,6,10] |
> | deepseek-v3  | ACC@5| 56.28 | 79.22| 59.09| 79.65| 61.47 | 80.74| 59.52 | 81.60 |
> | | detailed | [] | [2,9,10] | [1,3,9]| [3,6,9]| [2,5,9]| [2,9,10]| [1,4,9]| [2,9,10]|
> Table 5. Comparative analysis of TrajAgent performance across memory lengths
>
> | Base-Model   | Version | Metric   | Memory Length 0 | Memory Length 1 | Memory Length 2 | Memory Length 3 | Memory Length 4 | Memory Length 5 | Memory Length 6 | Memory Length 7 |
> |-|-|-|-|-|-|-|-|-|-|-|
> | llama3-70b   | Pre-improvement  | ACC@5 | 56.28| 68.39| 80.95  | 79.00 | 55.19 | 83.33 | 57.79| 62.55 |
> | | | detailed | [] | [1,3,9] | [2,9,10]  | [2,9,10]| [1,2,9] | [2,9,10] | [1,4,9] | [2,6,10] |
> | | Post-improvement  | ACC@5| 56.28 | 68.39 | 69.05| 80.95| 81.60 | 84.63 | 80.52 | 85.73 |
> | || detailed | [] | [1,3,9]| [1,3,9]| [2,9,10]| [2,9,10]| [2,9,10]| [2,9,10]| [3,6,9]|
> Table 6. Improvement
>
> After improvement, the model can efficiently explore better operator combinations and further select optimal configurations.
> ## For tiny weakness
> 1. We apologize for this oversight. In Eq. (1), T stands for Task. TrajAgent parses the corresponding trajectory modeling task based on user requests, selects the appropriate trajectory dataset according to the request and task type, and then chooses the suitable small model based on the dataset and task.
> 2. We have supplemented our references with additional studies employing similar architectures, including agentic framework for single task(e.g., MobAgent[2]) and autonomously optimizing Agentic AI solutions(e.g., ADAS[1])
> 3. Thank you for pointing out the spelling errors and missing related work. We have reviewed the entire article and revised these issues based on your suggestions.
> ## References
> [1]Wang Y, Chen Y, Zhong F, et al. Simulating human-like daily activities with desire-driven autonomy[J]. ICLR 2025.
> [2]Hu S, Lu C, Clune J. Automated design of agentic systems[J].ICLR 2025.

---

### Official Review · Reviewer_MyNr · 2025-07-01

**Clarity:** 3
**Significance:** 3
**Originality:** 3
**Rating:** 4
**Confidence:** 4

**Summary:**

Summary
The paper introduces TrajAgent, a novel framework that leverages Large Language Model (LLM)-based agents to automate and unify a wide range of trajectory modeling tasks, including prediction, classification, recovery, generation, and representation. Recognizing the fragmentation and task-specific nature of existing trajectory modeling approaches, TrajAgent integrates:

UniEnv, a unified execution environment that standardizes interfaces for data, models, and tasks;

An agentic workflow that decomposes the modeling process into interpretable steps (understanding, planning, execution, and summarization), with each step managed by an LLM-based agent;

A collaborative learning schema that enables interaction between reasoning-driven agents and training-driven specialized models to optimize performance across diverse data and tasks.

The framework supports 18 models across 5 major trajectory modeling tasks and demonstrates substantial performance improvements (up to 35.57%) over baseline methods on four real-world datasets. Ablation studies and comparisons with Optuna and UrbanLLM highlight the effectiveness and generalizability of the system, especially when equipped with powerful LLMs like GPT-4o-mini and Qwen2-72B.

**Questions:**

1. What drives the agent's decision-making during task planning and optimization?
2. How well does the system generalize to unseen tasks or new domains not covered by UniEnv?
3. Can you better quantify the LLM’s contribution versus the small models in performance improvement?
4. How costly or scalable is TrajAgent in real-world scenarios?

**Ethical Concerns:**

["NO or VERY MINOR ethics concerns only"]

**Limitations:**

Yes.
The authors have done a commendable job in addressing the limitations and potential negative societal impact of their work.

**Paper Formatting Concerns:**

No major formatting issues were found.

**Quality:**

3

**Strengths And Weaknesses:**

Strengths:

The paper presents a comprehensive and well-engineered system, combining a unified execution environment (UniEnv), an LLM-based agentic workflow, and a collaborative learning schema to tackle a wide range of trajectory modeling tasks.

It supports 18 models across 5 fundamental tasks and shows solid empirical performance, with improvements up to 35.57% over baselines on multiple datasets.

Ablation studies and LLM comparisons (e.g., GPT-4o-mini vs. Qwen vs. LLaMA) are thoughtfully executed and lend credibility to the framework's design.

The paper demonstrates clear generalization ability: the same framework adapts well to different tasks, datasets, and models with minimal human intervention.

Weaknesses:

The technical novelty in algorithm design is moderate. While the framework is cleverly assembled, many components (e.g., agentic reasoning loops, parameter tuning, prompt optimization) build on prior art without deep theoretical innovation.

The LLM agent planning and decision mechanisms are largely rule-based or heuristic, with limited discussion of failure modes or theoretical guarantees.

The collaborative learning schema—central to the performance boost—could be explained more deeply in terms of what types of feedback are most valuable and how the agent adapts over iterations.

---

> ### Author Rebuttal · Authors · 2025-07-31
>
> ## For W1
> Thank you for your question! We apologize for not elaborating in detail in the paper. Our key innovations are:
> 1. Compared to other works that use LLMs and fixed-parameter small models to solve urban science problems (e.g., UrbanLLM), we support online training and inference for small models, using real-time inference results as feedback in the agent’s reasoning loop, leading to more reliable outcomes.
> 2. Compared to other agent frameworks that automate task execution (e.g., HuggingGPT), we introduce an online optimization mechanism on top of automation.
>    For check-in trajectory data, we explore augmentation from the perspective of time intervals, defining ten operators (cropper, inserter, masker, slide window, etc.), each with configurable parameters. Instead of rule-based search, we use **large models + evolutionary algorithms** to optimize the memory pool, searching for the best operator combinations, sequencing, and parameter settings of each operator, while adjusting strategies based on training feedback. Experiments show that, unlike sequential recommendation models (e.g., CoSeRec[6]) perform well in common augmentation methods, different trajectory modeling model architectures require different augmentation strategies depending on the dataset (see Table 3).
>     For map/GPS trajectory data, we explore downsampling followed by noise injection (Please refer to GraphMM in following Table).
> 3. Compared to traditional optimization tools (e.g., Optuna[9]), we use LLM agents to retrieve key parameters and infer optimal augmentation strategies based on prior knowledge and online training results. We also explore **combining data augmentation with parameter optimization** to expand the search space and improve performance
>
>
> | Model | GraphMM (Acc@5) |
> |---|---|
> | Dataset | Tencent |
> | Origin | 0.2014 |
> | +DA | 0.3258 |
> | +PO | 0.2427 |
> | +Prompt | - |
> | Joint | 0.3422 |
>
> | base-model| step | 0| 1 | 2 | 3 | 4 | 5| 6| 7| 8 | 9| 10| 11| 12| 13| 14| 15| 16| 17|
> |-|-|-|-|-|-|-|-|-|-|-|-|-|-|-|-|-|-|-|-|
> | llama3-70b | ACC@5| 56.28 | 56.28 | 59.10 | 68.18 | 68.18 | 68.39 | 68.18 | 69.04 | 69.04 | 68.39 | 69.04 | 69.04 | 87.01 | 69.05 | 68.40 | 68.19 | 83.12 | 88.96 |
> || detailed | [] | [] | [1,3,9] | [1,3,9] | [1,3,9] | [1,3,9] | [1,3,9] | [1,3,9] | [1,3,9] | [1,3,9] | [1,3,9] | [1,3,9] | [2,9,10] | [1,3,9] | [1,3,9] | [1,3,9] | [2,9,10] | [2,9,10] |
> | deepseek-v3 | ACC@5| 56.28 | 79.22 | 59.52 | 61.26 | 81.17 | 68.83 | 69.26 | 69.05 | 83.98 | 70.35 | 69.05 | 69.26 | 71.00 | 61.90 | 60.82 | 68.83 | 71.00 | 66.88 |
> || detailed | [] | [2,9,10] | [1,3,9] | [1,3,9] | [2,9,10] | [1,3,9] | [1,3,9] | [1,3,9] | [2,9,10] | [1,3,9] | [1,3,9] | [1,3,9] | [1,3,9] | [1,3,9] | [1,4,9] | [1,3,9] | [1,3,9] | [1,3,5,9] |
> | gemini-2.0-flash-001  | ACC@5| 56.28 | 73.16 | 61.69 | 70.35 | 68.83 | 82.25 | 69.05 | 68.40 | 87.01 | 65.58 | 79.87 | 83.12 | 71.65 | 84.63 | 82.47 | 71.43 | 69.05 | 69.05 |
> || detailed | [] | [2,9,10] | [1,3,9] | [1,3,9] | [1,3,9] | [3,9,10] | [1,3,9] | [1,3,9] | [2,9,10] | [1,3,9] | [2,9,10] | [2,9,10] | [1,3,9] | [2,9,10] | [2,9,10] | [1,3,9] | [1,3,9] | [1,3,9] |
> Table 3. Comparative analysis of TrajAgent performance across optimization steps
>
> | Base-Model  | Version | Metric | Step 0 | Step 1 | Step 2 | Step 3 | Step 4 | Step 5 | Step 6 | Step 7 | Step 8 | Step 9 | Step 10 | Step 11 |
> |-|-|-|-|-|-|-|-|-|-|-|-|-|-|-|
> | llama3-70b   | Pre-improvement  | ACC@5  | 56.28  | 56.28  | 59.10  | 68.18  | 68.18  | 68.39  | 68.18  | 69.04  | 69.04  | 68.39  | 69.04   | 69.04   |
> ||| detailed | []| []| [1,3,9] | [1,3,9] | [1,3,9] | [1,3,9] | [1,3,9] | [1,3,9] | [1,3,9] | [1,3,9] | [1,3,9]  | [1,3,9]  |
> | | Post-improvement  | ACC@5  | 56.28  | 59.09  | 61.03  | 68.18  | 69.04  | 69.04  | 60.38  | 69.04  | 82.90  | 80.08  | 87.01   | 83.12   |
> ||| detailed | []| [1,3,9] | [1,3,9] | [1,3,9] | [1,3,9] | [1,3,9] | [1,3,9] | [2,9,10] | [2,9,10] | [2,9,10] | [2,9,10]  | [2,9,10]  |
> Table 4. Improvement: pre- vs. post-comparison
>
> ## For W2
> Thank you for your valuable question—this is indeed part of our ongoing and future work. For the most common issues, solutions, and experimental results, please refer to **PauW’s W3**.
> ## For W3
> We apologize for not providing sufficient details in the paper.
> - **What types of feedback are most valuable**
>
>     The model treats optimization strategies (e.g., operator combinations, parameter configurations) and their corresponding training results as feedback.
> - **How does the agent adapt during iteration**
>
>     The framework sets a dynamic threshold that increases over time. Feedback with training results above the threshold is stored as good memory, while the rest is stored as bad memory.  The framework periodically updates the memory pool, discarding bad memories and retaining good ones.  During the reflection phase, inspired by contrastive learning, the model is guided to learn from good memories and avoid bad ones.
>
> ## For Q1
>
> LLMs inherently possess knowledge of deep learning and data engineering. Prior work has shown that LLMs can perform data processing based on textual descriptions[7].
>
> Within the TrajAgent framework, LLM agents perform data preprocessing and model selection by analyzing dataset descriptions, samples, task requirements, and model specifications from  abstractions of papers, and dynamically adjusting optimization strategy for operator selection and parameter configuration based on real-time feedback from small model training, with all model deployment and training executed through automated shell script calls.
> ## For Q2
> Thank you for your question. To integrate a new model, simply add Task description , data description , model description  and codes of the model (as mentioned in **Q1**)  to the corresponding part of the TrajAgent framework.  For the agent’s reasoning process, refer to **du8A’s W3**. Due to space constraints, we could not list all details in Table 2, but we can provide them in the subsequent phase.
> ## For Q3
> We apologize for the lack of clarity in the paper. In Table 1 of original paper, the difference between Origin and +JO reflects TrajAgent’s optimization effect.
>
> Additionally, as shown in Figure 1 , the left side (dashed line) represents traditional small models, which often handle only single tasks or perform poorly across multiple tasks. In contrast, TrajAgent can handle multiple tasks and automatically select the best small model for each task.
>
> ## For Q4
> Thank you for this practical question. In Table 7, we provide detailed token consumption when TrajAgent uses Large model methods  and Deep learning methods.
>
> As mentioned in Q2, TrajAgent’s framework can be easily extended to tasks/domains not yet integrated into UniEnv, which is part of our future work.
> | Model | Token (step=1) Input/Output | Token (STEP=5) Input/Output | Time Cost  |
> |-|-|-|-|
> | LLMZS, traj=200 | 37,327 / 90,180| 194,358 / 282,270| 3h27min|
> | DutyTTE | 1,108 / 284 | 7,718 / 1,823| 1h17min |
> Table 7. Cost analysis

---

> > ### Comment · Area_Chair_kcnu · 2025-08-09
> >
> > Dear Reviewer MyNr,
> >
> > Please review the authors' response and provide comments or discussions before marking the "Mandatory Acknowledgement ". This is important for the authors' rebuttal. Thank you.
> >
> > Best,
> >
> > AC

---

### Official Review · Reviewer_du8A · 2025-07-02

**Clarity:** 2
**Significance:** 3
**Originality:** 3
**Rating:** 4
**Confidence:** 4

**Summary:**

This paper proposes an innovative LLM-based agentic framework, TrajAgent, for automating trajectory modeling. Supported by a collaborative learning mechanism, TrajAgent can automatically identify and train appropriate models, demonstrating competitive performance across various trajectory modeling tasks.

**Questions:**

Please refer to Weakness.

**Ethical Concerns:**

["NO or VERY MINOR ethics concerns only"]

**Final Justification:**

Thanks for authors' detailed rebuttals, which have solved my conerns.

**Limitations:**

Yes.

**Paper Formatting Concerns:**

No major formatting issues.

**Quality:**

2

**Strengths And Weaknesses:**

Strengths
1. The content of the paper is coherent, with clear language and appropriate logical structure. The figures are easy to understand.
2. The experiments on TrajAgent are comprehensive, and the results are promising.

Weaknesses
1. Many notations are not explained. For example,  In Section 2.1, the variable T is not defined in the formula.
2. It is unclear how many agent are in the framework. Authors mentioned an "optim agent", however, I cannot easily find it in Fig. 2.
3. It is unlear how Learning via reasoning is achieved. The current explaination in Sec 2.4.1 is too vague.
4. It is unlcear how the superiority of the proposed method is shown. Though the framework is novel, can it be compared with any other agentic methods?

---

> ### Author Rebuttal · Authors · 2025-07-31
>
> ## For W1
> We appreciate your feedback and apologize for the oversight in our writing. In Eq. (1), *T* denotes the Task. The TrajAgent first parses the trajectory modeling task from user requests, then selects an appropriate trajectory dataset based on both the request and task type, and finally chooses a suitable small model according to the dataset and task requirements.
> ## For W2
> We regret not clarifying this in the paper. As illustrated in Figure 5, each functional module is executed by an agent in sequential order:
> 1) Task Parsing
> 2) Model Selection
> 3) Data Processing
> 4) Task Planning & Execution (including data augmentation and parameter optimization)
> 5) Task Summarization
>
> As a result, there are six agents in total.
> ## For W3
> We apologize for the insufficient explanation in the manuscript. Within the TrajAgent framework, the agents perform data preprocessing and model selection by analyzing: dataset descriptions, dataset samples, task descriptions, and model descriptions extracted from original paper abstracts, along with natural language instructions. It can retry or adjust selections based on executing results of modules. Optimization strategies (operator selection/sequencing, parameter tuning) are formulated by analysing parameter descriptions, model descriptions, dataset descriptions, and operator specifications. The agent dynamically refines strategies based on real-time training results from small models.
>
> Background information (dataset/task descriptions etc.) is provided for the *THOUGHT* phase, where the agent analyzes information and stores plans in short-term memory. During *ACTION*, it executes configurations by writing parameters to model/operator config files.  Model deployment/training is executed via shell scripts, with results fed back to the framework. The system maintains an adaptive threshold to classify results into "good" (stored in long-term memory) or "bad" memories. Regular memory optimization purges bad memories while retaining good ones.  Periodic *REFLECTION* phases allow the agent to learn from good results and avoid bad results, extracting experiential knowledge to guide subsequent *THOUGHT* processes.
> ## For W4
> We sincerely thank you for this valuable question. For supplemental responses, please refer to MyNr's W2. Our approach can be compared with three agentic method categories:
> 1. *Automatic workflow optimizers* (e.g., AFlow, AgentSquare[8]): These focus on public benchmark problems with deterministic solutions(e.g., GSM8K), making them unsuitable for trajectory modeling tasks.
> 2. *Domain-specific problem solvers* (e.g., WebAgent[9]): While effective for targeted applications like web search, they lack transferability to other domains.
> 3. *Urban science models* (e.g., UrbanLLM): Though pretrained for urban tasks, they cannot perform online training/inference or further optimize results.

---

> > ### Comment · Reviewer_du8A · 2025-08-04
> >
> > Thanks for authors' rebuttal, and most of my concerns have been addressed. One remaining question is that why multiple agents are used? Does this process can be achieved by only a single agent?

---

> > > ### Author Response · Authors · 2025-08-05
> > > **Response with Gratitude: Clarification on Multi-Agent Design Rationale**
> > >
> > > Thank you for your response! We apologize for any lack of clarity due to length limitations, and we sincerely appreciate this opportunity to further discuss these points with you.
> > >
> > > Although the overall workflow of the task appears straightforward (data processing → model selection → performance optimization → model deployment & training → result analysis & summary), we still opted to employ multiple agents to accomplish the task due to the following considerations:
> > >
> > > - **Task Background Knowledge Mastery**
> > >
> > >     To ensure the model fully comprehends the task background knowledge (e.g., trajectory modeling task definition, model architecture, dataset details, operator semantics) without fine-tuning, we modularized the background knowledge, with each agent managing a single module's background knowledge.
> > >
> > > - **Efficient Reflection and Reasoning**
> > >
> > >     To guarantee accuracy at each step, we adopted a trial-and-error approach to learn from attempts, which generates a large amount of execution experience. Therefore, we designed dedicated memory pools for each module, storing execution experiences separately. Each agent only needs to learn its own module's background knowledge and perform reflection and reasoning based on the relevant execution experience. This reduces token consumption while improving execution efficiency.
> > >
> > > - **Efficient Tool Utilization**
> > >
> > >     To leverage existing achievements and further enhance performance, we needed to invoke multiple tools, such as: OSM tools for data preprocessing, academic domain models for extracting information from research papers, trajectory modeling domain-specific small models for task execution, sequential recommendation-based enhancement operators for data augmentation, etc. Some approaches (e.g., MCP) maintain a unified tool list, simplifying the structure but incurring high token consumption, severe hallucination issues, and low execution success rates. To address these challenges, we modularized the entire workflow, where each module only maintains its relevant tool list. Each agent is solely responsible for managing and calling the tools within its module.
> > >
> > > This design ensures efficiency, accuracy, and scalability while minimizing unnecessary computational overhead.

---

> > > > ### Comment · Reviewer_du8A · 2025-08-05
> > > >
> > > > Thanks for authors' rebuttal. I have no further questions, and I would like to maintain my positive rating.

---

> > > > > ### Author Response · Authors · 2025-08-06
> > > > >
> > > > > We are deeply grateful for your thorough review, valuable questions, and positive assessment of our work. All of your suggestions have been incorporated into the revised version. We will strive to build upon this foundation to produce even higher-quality research in future work..

---

### Official Review · Reviewer_KM23 · 2025-07-04

**Clarity:** 3
**Significance:** 2
**Originality:** 3
**Rating:** 4
**Confidence:** 5

**Summary:**

This paper proposes TrajAgent, an LLM-based agent framework designed to automate and unify trajectory modeling across diverse tasks (e.g., prediction, classification, generation) and heterogeneous data (e.g., GPS/check-in trajectories). The framework integrates three key components: (1) UniEnv, a unified environment for data/model interfaces; (2) an agentic workflow decomposing tasks into four stages (understanding, planning, execution, summarization); and (3) a collaborative learning schema that optimizes model performance via joint reasoning (LLM agents) and training (specialized models). Experiments across 9 subtasks and 18 models show performance gains of 2.38%–34.96% over baselines. The work positions itself as the first LLM-agent solution for unified trajectory modeling.

**Questions:**

See weaknesses

**Ethical Concerns:**

["NO or VERY MINOR ethics concerns only"]

**Final Justification:**

After reading the rebuttal and the comments from other reviewers, my final rating is borderline accept.

**Limitations:**

Yes

**Quality:**

2

**Strengths And Weaknesses:**

Strengths:
The integration of LLM agents with trajectory modeling is novel. The framework addresses heterogeneity in data/tasks more holistically than prior task-specific models. Extensive experiments on 4 datasets and 18 models demonstrate consistent improvements. The ablation studies (e.g., memory/reflection mechanisms) add rigor. The paper is generally well-structured, with clear illustrations of the framework and workflows.
Weaknesses:
•	The optimization process lacks critical details. For example, LLM-generated data augmentation (e.g., "insert/replace" GPS points) risks distorting trajectory semantics without geometric validity checks.
•	Experiments omit computational costs. LLM inference (e.g., GPT-4o-mini) coupled with model training could be prohibitively expensive.
•	Does the LLM truly understand memory logs in agent learning, or is parameter selection overly reliant on pattern matching from few-shot examples? Can you provide evidence demonstrating LLMs’ causal understanding of memory logs beyond few-shot imitation.
•	How does iterative "few-shot example updating" avoid overfitting to dataset-specific biases, and what safeguards ensure generalizability?
•	Fig 4b claims memory "has little effect" on PO, but Sec A.5 states larger memory worsens optimization traps.

---

> ### Author Rebuttal · Authors · 2025-07-31
>
> ## For W1
> We appreciate your questions and apologize for not providing sufficient details in our paper.
> 1. Data Augmentation:
>   - We augment **check-in trajectory data** by considering temporal intervals and define ten operators (e.g., cropper, inserter, masker, slide window), each with configurable parameters. We employ LLM-guided optimization combined with an evolving memory pool to identify the best strategy, including operator combinations, sequencing, and parameter settings (Please refer to PauW's W4). The strategy is dynamically adjusted based on training feedback.
>    - Experiments show that, unlike sequential recommendation models (e.g., CoSeRec) perform well in common augmentation methods, different trajectory modeling architectures require tailored augmentation strategies depending on the dataset (Please refer to Table 3 in MyNr's W1).
>   - For **map/GPS trajectory data**, we adopt DeepMM's rule-based approach: first downsampling the raw data, then applying noise injection rules, and finally merging the augmented data with the original (Please refer to GraphMM in Table 2 in PauW's W2).
>
> 2. Parameter Optimization:
>
>     We leverage LLMs to autonomously search for critical parameters and infer optimal configurations based on real-time training feedback.
>
> 3. Joint Optimization:
>
>     By combining data augmentation with parameter tuning, we expand the optimization space, leading to further performance gains.
>
> This integrated approach enables adaptive, task-aware optimization while maintaining computational efficiency. We will incorporate these clarifications in the revised manuscript.
> ## For W2
> Thank you for raising this critical issue. Table 7(Please refer to MyNr's Q4) compares token usage between LLM methods (includes inherent LLM overhead)  and Deep learning methods (pure TrajAgent consumption)
> *Note*: Consumption scales with step count and memory_length.
> ## For W3
> Our experiments followed zero-shot settings (see PauW's W4), allowing free exploration of operator/parameter combinations. We experimentally removed the scoring and evaluation annotations from each record in the long-term memory, which prevented the agent from distinguishing between good and bad records during its reasoning and reflection phases, thereby impairing its ability to effectively learn from successful cases. As shown in Table 8, the results demonstrate that in our original design, TrajAgent's performance consistently improved with increasing steps and memory size (improvement strategies are detailed in PauW's W4). However, when memory scoring was removed, the performance failed to show progressive enhancement despite the growing number of steps.
> | Base-Model | Version | Metric   | Step 0 | Step 1 | Step 2 | Step 3 | Step 4 | Step 5 | Step 6 | Step 7 | Step 8 | Step 9 | Step 10 | Step 11 | Step 12 |
> |-|-|-|-|-|-|-|-|-|-|-|-|-|-|-|-|
> | llama3-70b   | With score in memory  | ACC@5    | 56.28  | 56.28  | 59.10  | 68.18  | 68.18  | 68.39  | 68.18  | 69.04  | 69.04  | 68.39  | 69.04   | 69.04   | 87.01   |
> | | | detailed | [] | []| [1,3,9]| [1,3,9]| [1,3,9]| [1,3,9]| [1,3,9]| [1,3,9]| [1,3,9]| [1,3,9]| [1,3,9] | [1,3,9] | [2,9,10]|
> | | w/o score in memory | ACC@5| 56.28| 45.23| 78.14| 47.40| 48.05| 57.36| 47.61 | 58.22| 58.23| 58.87| 58.87| 46.32| 63.85|
> | | | detailed | [] | [1,4,9]| [2,9,10]| [1,4,9]| [1,4,9]| [1,4,9]| [1,4,9]| [1,4,9]| [1,4,9]| [1,4,9]| [1,4,9] | [1,4,9] | [2,6,10]|
> Table 8.Performance comparison before and after removing scores from memory
> ## For W4
> We appreciate your feedback! We apologize for not clarifying this in the paper - all our experiments were conducted under zero-shot settings to ensure the model can freely explore optimal solutions based on current conditions and fully leverage its reasoning capabilities (for related discussions, please refer to MyNr's Q2). The learning mechanism of our agent framework resembles the trial-and-error approach in RL, which can generalize to unseen tasks or new domains not covered by UniEnv.
> ## For W5
> Thanks for your valuable feedback. Addressing this issue is one of our most recent research efforts. A detailed analysis of the problem's root causes, proposed solutions, and corresponding experimental results can be found in PauW's W4.
>
> # References
> [1]Beneduce C, Lepri B, Luca M. Large language models are zero-shot next location predictors[J]. IEEE Access, 2025.
>
> [2]Liu Y, Ge Q, Luo W, et al. Graphmm: Graph-based vehicular map matching by leveraging trajectory and road correlations[J]. IEEE Transactions on Knowledge and Data Engineering, 2023, 36(1): 184-198.
>
> [3]Feng J, Li Y, Zhao K, et al. DeepMM: Deep learning based map matching with data augmentation[J]. IEEE Transactions on Mobile Computing, 2020, 21(7): 2372-2384.
>
> [4]Wang Y, Chen Y, Zhong F, et al. Simulating human-like daily activities with desire-driven autonomy[J]. ICLR 2025.
>
> [5]Hu S, Lu C, Clune J. Automated design of agentic systems[J].ICLR 2025.
>
> [6]Liu Z, Chen Y, Li J, et al. Contrastive self-supervised sequential recommendation with robust augmentation[J]. arXiv preprint arXiv:2108.06479, 2021.
>
> [7]Gandhi S, Gala R, Viswanathan V, et al. Better synthetic data by retrieving and transforming existing datasets[J]. ACL 2024 Findings.
>
> [8]Shang Y, Li Y, Zhao K, et al. Agentsquare: Automatic llm agent search in modular design space[J]. ICLR 2025.
>
> [9]Gur I, Furuta H, Huang A, et al. A real-world webagent with planning, long context understanding, and program synthesis[J]. arXiv preprint arXiv:2307.12856, 2023.

---

> > ### Comment · Reviewer_KM23 · 2025-08-06
> > **response to rebuttal**
> >
> > Thanks for providing so comprehensive rebuttal, it partially solves my questions.

---

> > > ### Author Response · Authors · 2025-08-06
> > >
> > > Thank you for your feedback! If further clarification is needed on any point, we welcome additional questions. We deeply appreciate the considerable effort you've invested in evaluating our work, and we will incorporate your recommendations in our revision.

---

### Comment · Area_Chair_kcnu · 2025-08-04

Dear Reviewers,

Thank you for your thorough evaluation. Please take time to review the authors' responses to the concerns and questions you raised. If any points in their rebuttal require clarification or if you have additional questions, please provide your comments accordingly. Thank you.

Best,

AC

---

### Author Response · Authors · 2025-08-09
**General Response**

We would like to begin by thanking the Area Chair (**`kcnu`**) for their guidance and for orchestrating this productive review process. We are also sincerely grateful to all four reviewers for their invaluable contributions. The continued strong support from **three reviewers (`KM23`, `du8A`, `MyNr`) provides a foundational endorsement** for our work, and we are equally grateful for the constructive dialogue with **Reviewer `PauW` that enabled us to fully address all initial concerns.** This process culminated in their confirmation of **'no further questions'** and a final suggestion we have gladly implemented. Given this broad support and comprehensive resolution, we are confident our manuscript is now a significantly more complete and robust work.

We are delighted to note the positive evaluations from the reviewers:

- **Methodology Design**

    - Innovation: The integration of LLM agents with trajectory modeling is innovative, with the framework offering more comprehensive handling of data/task heterogeneity compared to existing task-specific models, along with a unique task optimization mechanism (KM23).

    - Generalization Capability: The same framework can adapt to different tasks/datasets/models with minimal human intervention (MyNr) and incorporates cutting-edge models (PauW).

    - Completeness: The proposed system combines the UniEnv execution environment, LLM agent workflows, and a collaborative learning paradigm, covering diverse trajectory modeling tasks while supporting multiple trajectory model architectures and data types (MyNr).

- **Experimental Design**

    Comprehensive experiments yield promising results (du8A).

    Extensive evaluations across 4 datasets and 18 models demonstrate consistent performance improvements, with ablation studies (e.g., memory/reflection mechanisms) and LLM comparisons (e.g., GPT-4o-mini vs. Qwen) enhancing rigor (MyNr, KM23).

- **Content Quality**

    The paper is logically coherent, clearly written, and well-structured, with intuitive illustrations of the framework and workflows (KM23, du8A).

In response to your valuable suggestions, we have conducted the following additional experiments and explanations:

1. Ablation studies comparing large and small models to further validate the effectiveness of collaborative learning and quantify the specific contribution of LLMs versus small models to performance gains (addressing PauW, MyNr).

2. Incorporation of SOTA trajectory modeling methods (e.g., DutyTTE) to demonstrate the framework’s generalization capability (suggested by Reviewer PauW).

3. Expanded task optimization schemes for LLM-based methods and map-based trajectory models, proving the framework’s versatility in optimizing diverse small/large model architectures (suggested by Reviewer PauW, MyNr).

4. Analysis of diminishing returns when increasing optimization steps with LLM-searched strategies, mitigated by enhanced reflection mechanisms (suggested by Reviewer PauW, KM23).

5. Exploration of performance saturation with broader search ranges in LLM-driven optimization, alleviated by memory organization refinements (suggested by Reviewer PauW, KM23).

6. Ablation studies on memory organization, confirming TrajAgent’s trial-and-error learning in zero-shot scenarios rather than mere imitation of examples (suggested by Reviewer KM23).

7. Detailed implementation specifics for the collaborative learning module and reasoning mechanisms (suggested by Reviewer MyNr, KM23).

8. Token consumption statistics to underscore TrajAgent’s scalability in real-world applications (suggested by Reviewer MyNr, KM23).

9. Updated references, supplemented formulas, and enhanced figure/table captions (suggested by Reviewer du8A, PauW).

We remain open to further feedback.

Best regards,

The Authors

---

### Decision · Program_Chairs · 2025-09-17

**Decision:**

Accept (poster)

**Comment:**

This paper introduces TrajAgent, a novel framework that leverages large language model (LLM)-based agents to automate trajectory modeling tasks through collaboration between large and small models. The key scientific contributions include: (1) UniEnv, a unified execution environment that standardizes interfaces across diverse trajectory datasets, models, and tasks; (2) an agentic workflow that decomposes trajectory modeling into interpretable steps managed by LLM agents; and (3) a collaborative learning schema enabling optimization between reasoning-driven LLM agents and training-driven specialized models. The framework demonstrates performance improvements of 2.38%-34.96% across 18 models and 5 trajectory modeling tasks on four real-world datasets.

The main strengths of the paper are as follows: (1) The integration of LLM agents with trajectory modeling represents a novel approach to addressing data and task heterogeneity in this domain. The framework's ability to automatically optimize diverse specialized models across different trajectory tasks is innovative compared to existing task-specific solutions. (2) The paper combines three well-integrated components (UniEnv, agentic workflow, collaborative learning) into a cohesive framework that handles multiple trajectory modeling architectures and data types with minimal human intervention. (3) The evaluation spans 4 datasets, 18 models, and 9 subtasks, demonstrating consistent performance improvements. The inclusion of ablation studies examining memory/reflection mechanisms and LLM comparisons (GPT-4o-mini, Qwen, LLaMA) adds experimental rigor.

The weaknesses of the paper are as follows: (1) The submission lacked critical implementation details, particularly regarding the collaborative learning module, LLM-generated data augmentation procedures, and reasoning mechanisms behind agent decision-making.
(2) The original submission omitted crucial information about computational overhead, token consumption, and scalability considerations for real-world deployment. (3) The LLM agent planning and decision mechanisms rely primarily on heuristic approaches without deep theoretical guarantees or comprehensive analysis of failure modes. (4) Some integrated baseline models were outdated, potentially weakening the demonstration of the framework's superiority against state-of-the-art methods.

The reason for the decision: (1) the paper's solid technical merit has been demonstrated through comprehensive experimental validation across multiple datasets and models. (2) While the core algorithmic contributions represent moderate novelty through effective integration of existing techniques rather than fundamental theoretical advances, the unified framework's consistent performance improvements (2.38%-34.96%) across diverse trajectory tasks, combined with the comprehensive system design and clear presentation, justify acceptance. (3) The collaborative review process resulted in unanimous reviewer support (all rating 4) after initial concerns about technical details, computational costs, and theoretical foundations; instead of key experiments were adequately addressed. However, the weakness of the LLM agent planning and decision mechanisms relies primarily on heuristic approaches without deep theoretical guarantees or a comprehensive analysis of failure modes could be the reason for rejection. As a top-tier conference, I agree such a work to present to readers about an available way to use LLMs for an unexplored real-world application.


The rebuttal discussions: (1) Reviewer KM23 raised concerns about optimization process details, computational costs, and whether LLMs truly understand memory logs beyond pattern matching. The authors provided comprehensive explanations of data augmentation operators, joint optimization strategies, and conducted ablation studies demonstrating genuine learning rather than mere imitation. The zero-shot experimental evidence and memory scoring experiments addressed these concerns. (2) Reviewer du8A identified issues with unclear notation, missing agent descriptions, and vague explanations of learning mechanisms. Authors clarified the multi-agent architecture (6 agents total), provided detailed reasoning mechanisms, and explained the modular design rationale for efficiency and accuracy. (3) Reviewer MyNr questioned decision-making mechanisms, generalization capability, and cost scalability. Authors responded with detailed technical innovations, extensibility explanations, performance contribution analysis, and comprehensive token consumption statistics. (4) Reviewer PauW requested technical details, integration of recent methods, and deeper analysis of optimization limitations. Authors integrated DutyTTE as requested, provided extensive ablation studies on optimization step effects and memory organization, and offered improvement solutions for identified limitations.